# New orphan disease therapies from the proteome of industrial plasma processing waste- a treatment for aceruloplasminemia

Alan Zanardi[1,11], Ilaria Nardini[2,11], Sara Raia[1], Antonio Conti[1], Barbara Ferrini[1], Patrizia D'Adamo [3], Enrica Gilberti[4], Giuseppe DePalma [4], Sara Belloli [5,6], Cristina Monterisi[5], Angela Coliva [5], Paolo Rainone [5,6,7], Rosa Maria Moresco[5,6,7], Filippo Mori[2], Giada Zurlo[2], Carla Scali[2], Letizia Natali[2], Annalisa Pancanti[2], Pierangelo Giovacchini[2], Giulio Magherini [2], Greta Tovani[2], Laura Salvini[8], Vittoria Cicaloni[8], Cristina Tinti[8], Laura Tinti[8], Daniele Lana [9], Giada Magni [10], Maria Grazia Giovannini [9], Alessandro Gringeri[2], Andrea Caricasole [2,12 ✉] & Massimo Alessio [1,12 ✉]

Plasma-derived therapeutic proteins are produced through an industrial fractionation process where proteins are purified from individual intermediates, some of which remain unused and are discarded. Relatively few plasma-derived proteins are exploited clinically, with most of available plasma being directed towards the manufacture of immunoglobulin and albumin. Although the plasma proteome provides opportunities to develop novel protein replacement therapies, particularly for rare diseases, the high cost of plasma together with small patient populations impact negatively on the development of plasma-derived orphan drugs. Enabling therapeutics development from unused plasma fractionation intermediates would therefore constitute a substantial innovation. To this objective, we characterized the proteome of unused plasma fractionation intermediates and prioritized proteins for their potential as new candidate therapies for human disease. We selected ceruloplasmin, a plasma ferroxidase, as a potential therapy for aceruloplasminemia, an adult-onset ultra-rare neurological disease caused by iron accumulation as a result of ceruloplasmin mutations. Intraperitoneally administered ceruloplasmin, purified from an unused plasma fractionation intermediate, was able to prevent neurological, hepatic and hematological phenotypes in ceruloplasmin-deficient mice. These data demonstrate the feasibility of transforming industrial waste plasma fraction into a raw material for manufacturing of new candidate proteins for replacement therapies, optimizing plasma use and reducing waste generation.

[1] Proteome Biochemistry, COSR-Centre for Omics Sciences, IRCCS Ospedale San Raffaele, Milano, Italy. [2] Research and Innovation, Kedrion S.p.A., Loc, Bolognana, Gallicano, Italy. [3] Mouse Behavior Facility, IRCCS Ospedale San Raffaele, Milano, Italy. [4] Unit of Occupational Health and Industrial Hygiene, Department of Medical and Surgical Specialties, Radiological Sciences and Public Health, University of Brescia, Brescia, Italy. [5] Nuclear Medicine and PET Cyclotron Unit, IRCCS Ospedale San Raffaele, Milano, Italy. [6] Institute of Molecular Bioimaging and Physiology-IBFM, CNR, Segrate, Italy. [7] Medicine and Surgery Department, University of Milano – Bicocca, Monza, Italy. [8] Toscana Life Sciences Foundation, Siena, Italy. [9] Department of Health Sciences, Section of Clinical Pharmacology and Oncology, University of Florence, Firenze, Italy. [10] Institute of Applied Physics "Nello Carrara", National Research Council (IFAC-CNR), Sesto Fiorentino, Italy. [11] These authors contributed equally: Alan Zanardi, Ilaria Nardini. [12] These authors jointly supervised this work: Andrea Caricasole, Massimo Alessio. ✉email: a.caricasole@kedrion.com; alessio.massimo@hsr.it

Human plasma contains a complex proteome with more than 2000 proteins surveyed[1,2], with great potential therapeutic and diagnostic interest, but remains still largely untapped. Indeed, at present plasma proteins are only minimally exploited, with about 20 blood products used as replacement/supplementation therapies in clinical practice[3–5] of which albumin (for critical care indications) and polyvalent immunoglobulin (for the treatment of primary immunodeficiencies and autoimmune disorders) represent >70% of the global output of plasma-derived therapeutics. Protein replacement therapy (PRT) represents a consolidated approach in the treatment of many rare and ultra-rare diseases where relevant proteins are hypofunctional or deficient[2,6]. Clinically validated examples are represented by therapies for coagulation defects (*e.g.*, FVIII or FIX concentrates for hemophilia A and B[7]), and primary and secondary immunodeficiencies[8,9]. However, more than 7,000 congenital rare diseases are presently known[6], and more than 90% of these remain without effective therapies[10]. A large number of these disorders are characterized by hypofunctionality or low levels of a single protein, whose replacement could restore the missing biological function[6]. When the disease is associated with a dysfunctional plasma protein, fresh frozen plasma is often administered if specific replacement therapy is not available[11]. However, when a relevant plasma-derived protein concentrate or a recombinant protein are available, these can instead replace the missing protein efficaciously and reduce the risks and inconvenience of repeated plasma administration, such as fluid overload and transfusion-related acute lung injury[12].

Plasma-derived protein therapeutics are purified through a process essentially developed more than 70 years ago[13], which involves the processing of human plasma pools into different fractions (intermediates), only some of which are used to produce the proteins of interest while others remain unused and are discarded as industrial waste (waste fractions). Human plasma is a precious resource and represents a very important proportion of the cost of a plasma-derived product. Production of a new plasma-derived therapy can divert plasma from the manufacture of existing products thus limiting their availability. The high cost of goods together with the small patient populations can make plasma-derived therapies for rare diseases very expensive, and even uneconomical to produce in the case of ultra-rare conditions. Enabling protein purification from waste fractions rather than whole plasma substantially improves the economics of plasma-derived therapies development, particularly for ultra-rare conditions, and results in a more efficient and ethical use of blood and plasma donations. This requires knowledge and exploitation of the proteome of such unused intermediates. Although the unfractionated plasma proteome has been studied[3–5], information on proteins in plasma fractions discarded during the production of plasma products are sporadic, confined to plasma manufacturers, and poorly represented in the literature[14]. Clearly, not all proteins present in these waste fractions may represent useful and feasible targets for the development of PRTs. Additionally, their biochemical and biological properties may be altered upon purification from a plasma fractionation waste fraction, relative to the same protein purified from whole plasma. Therefore, a careful analysis and filtering of this proteome is required to identify and prioritize therapeutically relevant candidates, and integrity and activity of any selected candidates needs to be carefully determined. In this work, we have investigated and characterized the proteome of the discarded fractions from an industrial plasma fractionation plant, identifying hundreds of proteins through a mass spectrometry approach. The resulting proteome was subjected to a bioinformatics and data mining analysis with the aim to prioritize proteins based on therapeutic potential, unmet medical need, availability of translationally

relevant animal models and development feasibility, resulting in a subset of prioritized proteins which included novel PRT candidates as well as PRTs already available commercially. Of the proteins representing novel candidate PRTs, ceruloplasmin (CP), the most abundant plasma ferroxidase, stood out as a particularly interesting example because its deficiency characterizes aceruloplasminemia (ACP), an ultra-rare disease[15]. Its prevalence in Japan is $0.5:10^6$, with about 130 cases reported worldwide to date[15–18]. ACP is caused by mutations in the *CP* gene and results in iron accumulation in several tissues, including the brain where it leads to progressive and severe neurodegeneration[17]. In the absence of a standard treatment and of a specific ceruloplasmin replacement therapy, ACP is currently addressed with iron chelation therapy and fresh frozen plasma transfusion, which can only partially restore the levels of the missing CP protein[17–19]. Coherent with sporadic clinical findings on the benefit of plasma treatment in patients, CP replacement therapy of *cp* knock-out (cpKO) mice results in mitigation of the neurological and neurodegenerative phenotype together with the reduction of systemic dysregulation[20,21]. This evidence prompted us to evaluate CP as a candidate PRT from the plasma fractionation waste intermediate proteome to address iron-related neurodegeneration such as that observed in ACP in a cpKO mouse model.

## Results

### Shotgun proteomics and bioinformatics analysis for proteins prioritization.

A shotgun proteomics approach was performed to investigate the proteins present in the solubilized unused intermediates (Fraction I, FI; Fraction III, FIII and Fraction $IV_{1-4}$, $FIV_{1-4}$) from a major industrial plasma fractionation plant located in Italy (Fig. 1a). Only high confidence proteins previously characterized as plasma proteins (https://www.proteinatlas.org/) were considered, for a total of 166 proteins in FI, 268 in FIII and 138 in $FIV_{1-4}$ (Supplementary Data 1 and Fig. 1b). Within these, 308 human plasma proteins were found, of which 78 (ca. 25%) were present in all three unused intermediates and 186 (60%) in at least 2 intermediates (Fig. 1b). Unique proteins in the aggregated unused intermediate proteome ($N = 308$) were analyzed through a series of successive analytical filters (Fig. 1c), including public databases search to determine plasma concentration and biological function. Next, a disease association analysis[22] was performed to identify the therapeutic potential for each protein, with a focus on rare and monogenic orphan diseases. The association was considered reliable with a Gene-Disease Association (GDA) score, which reflects how well established is a particular association based on current knowledge, of 0.7 or higher[22]. With this cut-off, in FI we found 68 proteins associated with 115 disorders, in FIII 99 proteins linked to 165 diseases whereas 69 proteins and 107 diseases in $FIV_{1-4}$ (Supplementary Data 2). First, evidence of the therapeutic use of the identified proteins as licensed drugs was determined, as well as evidence of their assessment in clinical trials. When this was not found, we identified any literature evidence that a pertinent disease phenotype can be mitigated with delivery/expression of the protein of interest in relevant animal models. The result of this analysis, including the GDA Score from 0.7 to 1 and the related disease for each protein, is illustrated in Fig. 2a (for the full list see Supplementary Data 3, prioritized proteins). The stronger the gene-disease relationship, the higher is the GDA score. Examples are *F11* and Hereditary Factor XI Deficiency (score= 1.00), *F13A1* and Hereditary Factor XIII Deficiency (score= 0.98), *F9* and Hemophilia B (score= 1.00), *SERPINC1* and Antithrombin III Deficiency (score= 1.00), *VWF* and von Willebrand Disease, Type 2 (score= 1.00), *CP* and Ceruloplasmin Deficiency (score= 0.90), *HP* and Congenital haptoglobin deficiency (score = 1,00),

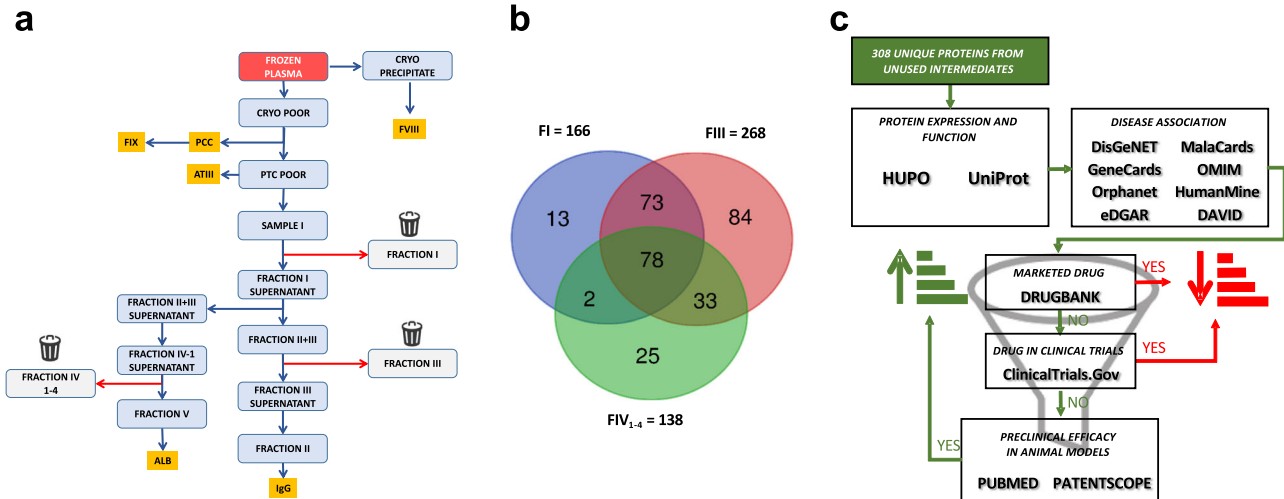

**Fig. 1 Proteomic analysis of industrial plasma fractionation intermediates. a** Simplified example of an industrial plasma fractionation process as applicable in the examined plasma fractionation manufacturing plant. Abbreviations used refer to Prothrombin Complex Concentrate (PCC), Coagulation factors VIII and FIX (FVIII and FIX), Antithrombin III (ATIII), Albumin (ALB) and Immunoglobulin (IgG). The entire process generates at least three unused intermediates (Fraction I, FI; Fraction III, FIII and Fraction IV$_{1-4}$, FIV$_{1-4}$). **b** Venn diagram of the proteomes of FI, FIII and FIV$_{1-4}$. For the full list of unique proteins see Supplementary Data 1 **c**. Scheme of the filtering process leading to the prioritization of unused intermediate proteins based on function and plasma proteome association, disease association, therapeutic value/potential and evidence of efficacy as protein replacement therapies. For the complete prioritization analysis see Supplementary Data 2.

*TF* and Congenital atransferrinemia (score= 0.91). To demonstrate feasibility of developing a PRT from unused fractionation intermediates, we focused on proteins for which preclinical PRT efficacy, obtained with a plasma-derived product, is already available (Fig. 2b) resulting in the prioritization of haptoglobin (HP), CP, transferrin (TF) and hemopexin (HPX). HPX was also included because of the robust evidence for preclinical efficacy in a hemopexin-unrelated rare disease, namely sickle cell disease[23] (see Supplementary Data 3, prioritized proteins). From these, one example was selected for progression to purification and characterization from unused plasma fractionation intermediates, for demonstration of efficacy of PRT in a translationally relevant mouse model. CP was selected for several reasons. First, the univocal association with a monogenic, ultra-rare disease which is currently treated with a therapy including plasma, to provide the missing CP protein[18]. Secondly, its stability in vivo ($T_{1/2}$ of 5 days[24]) and the availability of a translationally relevant mouse model[25]. Finally, CP was selected because of the existence of a proof-of-concept PRT study using a plasma-derived CP in a cpKO mouse model[20], which would enable us to at least indirectly compare the in vitro properties and in vivo efficacy of a CP purified from an unused plasma fractionation intermediate to a commercially available research grade CP purified from plasma.

**Ceruloplasmin purification and characterization from plasma fractionation unused intermediates**. As CP is present in the proteomes of all three unused plasma fractionation intermediates, these samples were biochemically characterized for CP content, integrity, purity, and activity to identify the most suitable starting material for its purification. Among the three intermediates, FIV$_{1-4}$ was found to contain CP protein with the highest integrity (as determined by electrophoretic pattern; Fig. 3a), oxidase activity, antigen content, and purity (Fig. 3b–d; Supplementary Data 4), and was therefore used for subsequent CP purification and characterization. To design a proper purification procedure for the isolation of CP from the other components of FIV$_{1-4}$, a quantitative characterization of the most abundant proteins in this intermediate (selected among the ones identified by proteomic analysis) was performed. Following an analysis of most

abundant contaminants (Supplementary Fig. 1), these were removed by chromatography resulting in a CP concentrate (hereafter defined as kCP) of high purity (80% by relative CP levels and 98% by proteomics analysis; see Supplementary Data 3, ceruloplasmin characterization; Fig. 3e), the difference likely reflecting the presence of CP fragments not detected by the CP antibody-based biochemical assay (Fig. 3j). By Western blotting analysis, CP purified from FIV$_{1-4}$ was wholly comparable to the commercially available plasma-derived CP protein employed in previous in vivo studies[20,21,26], and to CP detected in whole plasma (Fig. 3j, Supplementary Fig. 2, and Supplementary Fig. 3a)[27]. As detected by Western blotting, partial degradation is apparent in plasma, and to a higher extent in plasma fractions and purified plasma-derived CP samples (kCP and commercially available CP; e.g. see Supplementary Fig. 2a). This is likely caused by events occurring during industrial plasma fractionation rather than purification, as the partial proteolysis profile of the starting material (FIV$_{1-4}$) is identical to that of the purified CP product (kCP; Supplementary Fig. 2a, b). As Thrombin is reported as a protease that causes limited proteolysis of CP observed both in vitro, during purification and in vivo[28], this is likely a contributor of the observed partial degradation. The holoenzyme form of CP (which has six copper atoms buried in its folded, enzymatically active state), was correctly detected in kCP in a manner comparable to commercially available protein and to CP present in whole plasma[29] (Supplementary Figs. 2 and 3). kCP, when compared to the original intermediate, demonstrated at least a > 40-fold enrichment in purity (Fig. 3h; Supplementary Data 4) and a two-fold enrichment in CP protein levels (Fig. 3f; Supplementary Data 4) and oxidase activity (Fig. 3g; Supplementary Data 4), while preserving specific oxidase activity during the purification process (Fig. 3i; Supplementary Data 4). Accordingly, kCP demonstrated robust specific oxidase and ferroxidase activity (Figs. 3k and 3l; Supplementary Data 4).

**Administered kCP does not induce acute toxicity in mice and is detectable in the blood of cpKO mice**. kCP was tested for any potential acute toxicity in cpKO mice upon administration at 5 μg/g and 10 μg/g (the former is the dosage used previously in

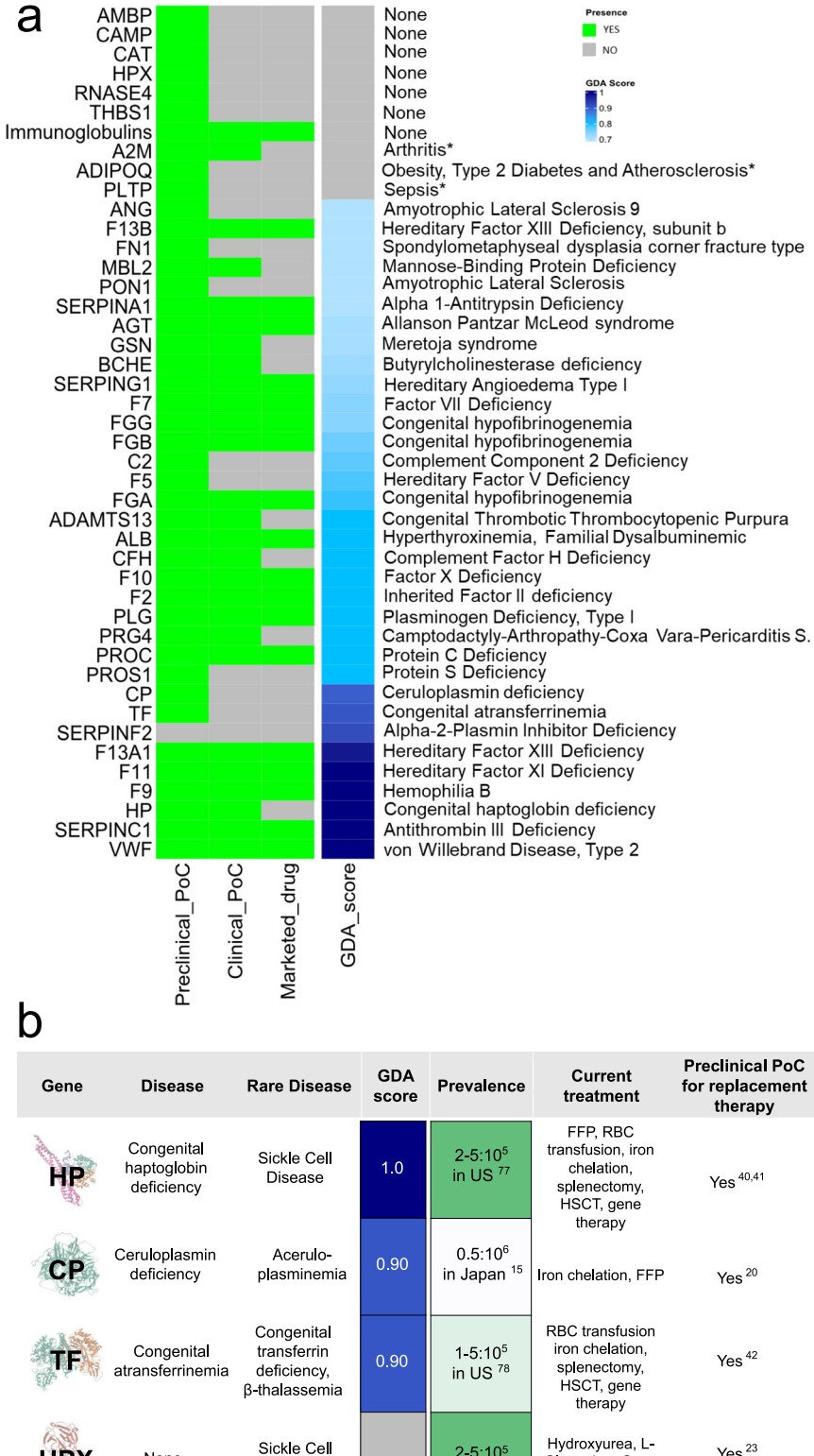

**Fig. 2 Selection of candidate protein replacement therapeutics from the unused intermediates proteome for proof-of-concept studies. a** Heatmap reporting prioritized proteins from the proteome of unused intermediates. For each protein the associated disease is indicated ("*"= non-rare diseases) and the Gene-Disease Association (GDA) from 0.7 (in light blue) to 1 (in navy blue). The higher is the score, the higher is the gene-disease association. **b** Plasma proteins present in unused fractionation intermediates for which evidence of preclinical efficacy is available. Applicable prevalence data are indicated[15,77,78]. Relevant references for Preclinical proof of concept (PoC) for replacement therapy for Haptoglobin[40,41] (HP), ceruloplasmin[20] (CP), Transferrin[42] (TF) and Hemopexin[23] (HPX) are indicated.

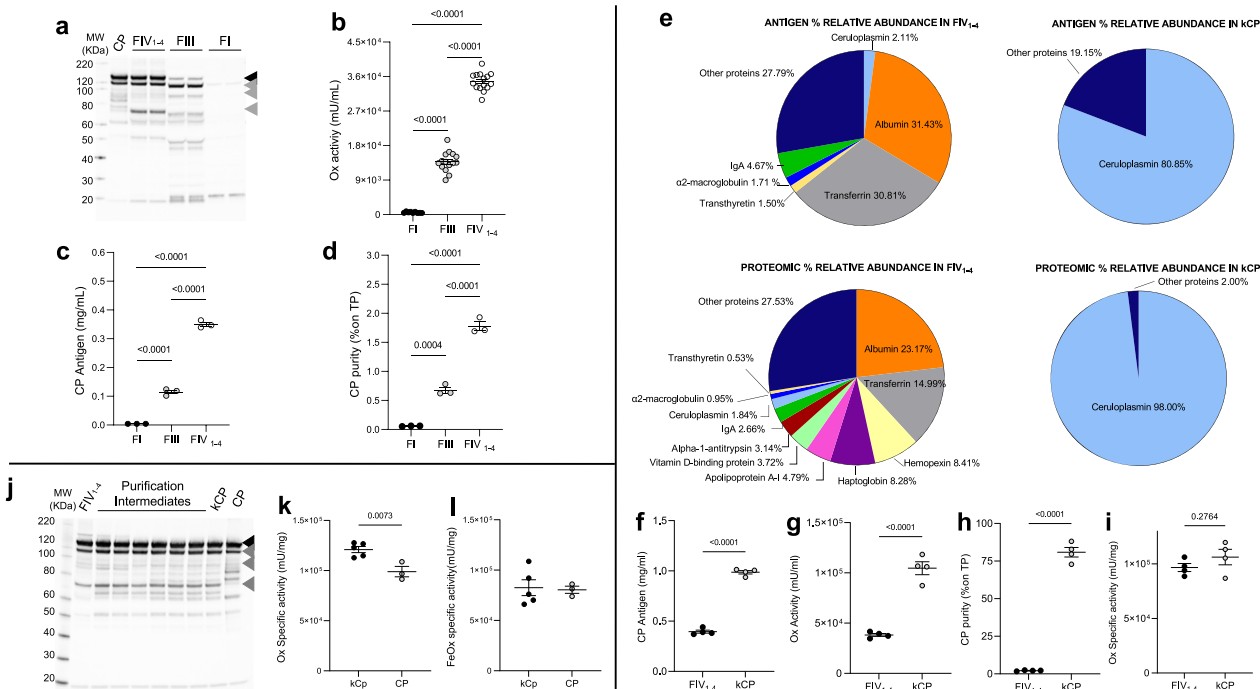

**Fig. 3 Selection of unused plasma industrial fractionation intermediates as source material and purification and characterization of CP from FIV$_{1-4}$.**
**a** FIV$_{1-4}$ is the most suitable starting material for CP purification. CP Immunoblotting analysis of CP in commercially available plasma-derived CP, FIV$_{1-4}$, FIII and FI. Intact CP has a predicted MW of 132 kDa (black arrowhead), and FIV$_{1-4}$ contains the highest proportion of intact protein. Known protease cleavage sites likely result in the production of smaller fragments of 70, 90 and 116 kDa respectively[28]. **b–d** FIV$_{1-4}$ contains the highest levels of CP enzymatic activity (oxidase assay, **b**), of CP protein **c**, and the highest CP levels relative to total protein **d**. **e–i** Biochemical characterization of CP purified from FIV$_{1-4}$ (kCP). **e** Relative protein composition in FIV$_{1-4}$ and in kCP as analyzed by biochemical methods (top panels) and mass spectrometry (bottom panels; see also Supplementary Data 3, ceruloplasmin characterization). **f–i** Comparison between FIV$_{1-4}$ and kCP. Enrichment in CP content **f**, oxidase activity **g** and purity **h** in kCP vs FIV$_{1-4}$. Specific oxidase activity is preserved during kCP purification **i**. **j** Immunoblotting illustrating preservation of CP content and integrity from the starting material (FIV$_{1-4}$), through purification intermediates and to the final product, kCP, with commercially available CP as a reference. **k, l** kCP concentrate specific activity as measured by both oxidase **k** and ferroxidase **l** assays. Data are presented as means ± SEM; each dot corresponds to one analysis in **a–d** and to one batch in **e–l**. Numerosity: $N = 9$ for FI, $N = 15$ for FIII and FIV$_{1-4}$ in **b**; $N = 3$ for FI, FIII and FIV$_{1-4}$ in **c, d**; $N = 4$, three analyses for each batch in panels **f-i**; $N = 3$ for CP, $N = 5$ for kCp, three analyses for each batch in **k** and **l**. Statistical $P$ values were evaluated by one-way ANOVA followed by Tukeys's multiple comparisons test in panels b-d and by unpaired t-test in panels f-i and k-l.

cpKO mice[20,21,26]. Animals were injected intraperitoneally twice (at time 0 and after 5 days) and were monitored until day 10. No signs of toxicity were observed as measured by overall appearance, gross behavior, motility, body weight and, at necroscopy, by macroscopic appearance and weight of relevant organs (Supplementary Figs. 4 and 5). Similarly, long term treatment in WT mice with kCP (5 µg/g intraperitoneallly administered to 6 months-old cpKO mice for 4 months every 5 days) did not provide evidence of toxicity in terms of weight, behavior, and organ appearance/weight (Supplementary Fig. 6). Western blot detection of intraperitoneally injected kCP (5 µg/g) indicated that, in cpKO mice, the level of circulating kCP reached a peak after 3 h and it was still detectable in the blood 24 h after administration (Supplementary Fig. 7a–c). The kinetics of kCP appearance in the blood of cpKO mice was mirrored by the recovery of ferroxidase activity (Supplementary Fig. 7d). Considering exclusively the kCP (human protein) signal, in WT mice blood CP levels showed similar kinetics to that observed in cpKO mice, but the signal intensity was somewhat lower (Supplementary Fig. 7c). This suggested that in WT mice the removal of kCP is faster than in cpKO mice, or that injected CP is more stable in the blood of cpKO mice. Therefore, reduced protein degradation or excretion might occur in the absence of the endogenous CP and/or injected kCP is more stable in the blood of cpKO mice due to the interactions with other plasma components (e.g., with myeloperoxidase, lactoferrin, transferrin, etc.; e.g. see[28] and references

therein), interactions that in the plasma of WT mice are not possible since they are already occurring with the endogenous CP. Similar evidence was obtained by evaluating the accumulation of radiolabelled $^{64}$Cu-kCP in the blood and plasma of cpKO and WT mice. Indeed, 21 h after administration a significantly higher accumulation of radiolabelled kCP was observed in cpKO mice compared to WT (Supplementary Fig. 8).

**kCP treatment induces partial rescue of both protein level and ferroxidase activity in the plasma of cpKO mice, and it elicits an anti-CP serological immune response unable to neutralize the CP ferroxidase activity.** kCP was administered intraperitoneally (5 µg/g) to 6 months-old cpKO mice for 4 months every 5 days, and the plasma level of kCP at the end of the treatment was evaluated. kCP was clearly detectable in treated cpKO mice (Fig. 4a). An average of 33 pg/µl of kCP was detected in the plasma of treated mice, which is still, on average, about 500-fold lower than the endogenous CP concentration in WT mice (15–20 ng/µl;[20]). Despite the low protein concentration found in the plasma of the mice at the end of the treatment, the rescue of ferroxidase activity in plasma of cpKO was significant (Fig. 4b). This was about 4% of the reference ferroxidase activity in WT mice (only about 25-fold lower than physiological levels; compare with 500-fold lower protein levels), confirming the reported observation that human CP displays higher enzymatic activity

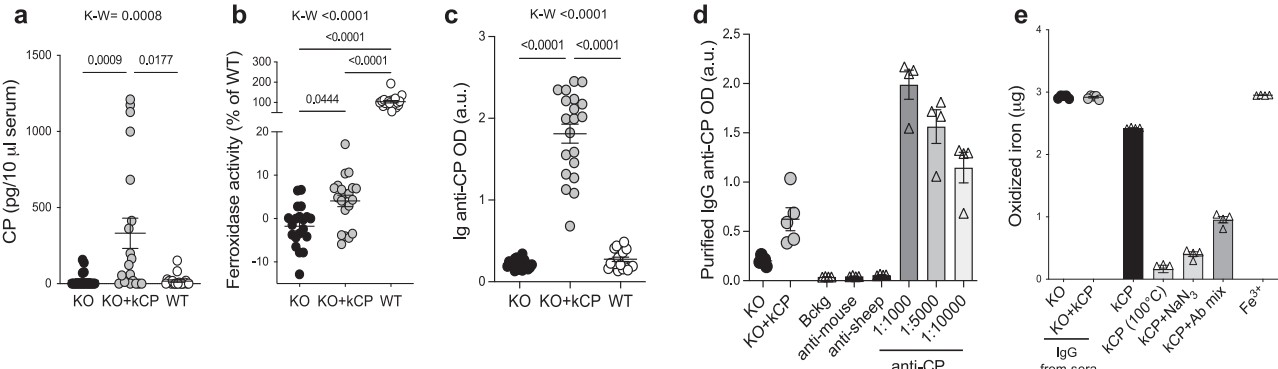

**Fig. 4 Treatment with kCP results in partial rescue of protein level and ferroxidase activity in the plasma of cpKO mice, and it elicits a non-neutralizing anti-CP serological immune response. a** ELISA evaluation of human CP in plasma of mice treated for 4 months either with kCP (KO+kCP) or saline (KO, WT). **b** Ferroxidase activity measured in the same samples as in **a**, expressed as percentage of the activity in WT mice. **c** ELISA detection of anti-CP IgG and **d** ELISA analysis of CP-binding capability of IgG purified from the serum of mice. **e** Ferroxidase activity evaluated on human kCP incubated with IgG purified from the serum of mice. Data are presented as means ± SEM, dots correspond to values from individual animals ($N = 20$ mice/group in **a**–**c**); in **d** and **e** each dot corresponds to a pool of 4 randomly selected sera from the same experimental group in KO and KO+kCP ($N = 5$ independent pools) and to $N = 4$ independent experiments for the control conditions (triangles). Statistical $P$ values were evaluated by Kruskal-Wallis (K-W) test followed by Dunn's **a**, **c** and uncorrected Dunn's **b** post-test analysis.

than mouse CP[30]. These results indicated that CP expression level and activity were partially but significantly recovered in the plasma of cpKO mice after 4 months of PRT with kCP.

A possible limitation of the PRT approach is the induction of an immune response against the injected protein. Therefore, at the end of kCP PRT we evaluated the presence of neutralizing anti-CP antibodies, which could affect therapeutic efficacy by limiting CP protein levels and/or ferroxidase activity. The injected kCP resulted to be immunogenic and, indeed, anti-CP antibodies were detected in cpKO mice after PRT (Fig. 4c). To investigate whether these antibodies were capable of neutralizing kCP, immunoglobulins G (IgG) have been purified from the plasma of cpKO and cpKO treated mice. The IgG purified from kCP-treated cpKO mice could recognize kCP (Fig. 4d) but were not able to neutralize kCP ferroxidase activity as, on the contrary, did a mix of commercial polyclonal anti-CP antibodies known to be neutralizing[20], CP-heat inactivation or treatment with sodium azide, an inhibitor of CP activity[20,30] (Fig. 4e). These results suggested that the anti-CP immune response induced in kCP treated mice did not affect the efficacy of PRT by limiting the recovery of CP-ferroxidase activity.

**kCP enters the brain of cpKO mice ameliorating motor coordination, reducing iron accumulation and preventing neurodegeneration.** The previously reported ability of administered CP to cross the brain–barrier-systems and to enter in the central nervous system (CNS) of cpKO mice[20,26] was confirmed by the ex-vivo analysis of the accumulation of $^{64}$Cu-labeled kCP in the brain of the mice. Indeed, the $^{64}$Cu-kCP showed a significantly higher accumulation in 10 months-old cpKO mice compared to WT after 21 h from the administration (Fig. 5a). The accumulation of radiolabelled-kCP in the brain of the animals confirmed the feasibility of the therapeutic approach to address the neurological symptoms. At 6 months of age no neurological symptoms associated to motor coordination were detectable in cpKO mice in neither rotarod nor grid test (Fig. 5b, c), as previously observed[20]. However, at 10 months of age cpKO mice demonstrated clear signs of motor coordination impairment compared to WT animals, on these tests, indicative of the onset of neurological symptoms (Fig. 5b, c), as described[20]. Therefore, motor coordination of the animals was assessed and the end of kCP PRT (5 µg/g kCP administered to 6 months-old cpKO mice for 4 months every 5 days) as

a symptomatic proxy of neurodegeneration. kCP-treated cpKO mice demonstrated a significant improvement of motor coordination compared to untreated cpKO mice, on both rotarod (where performance was completely restored to WT levels; Fig. 5b) and grid tests (where performance was significantly improved relative to the performance of cpKO mice; Fig. 5c). In addition, the beam walking test (used to examine gait of the mice in an environment that challenges their ability to balance themselves) was performed. In this motor test, cpKO mice performed significantly worse relative to WT mice, and kCP treatment restored coordination to that of WT mice (Fig. 5d). These results indicated that kCP PRT was able to robustly ameliorate neurological symptoms associated with motor coordination. At the end of PRT, significant kCP accumulation was detected in the brain of kCP-injected cpKO mice (Fig. 5e), coherent with the accumulation of radiolabelled kCP data (Fig. 5a) and with published evidence[20,26]. CP accumulated in variable amounts ranging from 0.35-6.7 pg/µg of total brain extract, corresponding to ca. 190- to 10-fold lower levels relative to endogenous CP estimated in the brain of WT mice (66.6 pg/µg of total brain extract)[20]. By inductively coupled plasma mass spectrometry analysis, we evaluated the total iron content in the brain homogenates of 10-month-old mice. Larger iron accumulation was found in cpKO mice compared to both WT mice and cpKO mice treated with kCP. CP-treatment significantly prevented iron deposition in the brain of cpKO mice, maintaining levels comparable to those found in WT mice (Fig. 5f). In cpKO mice significant iron accumulation was found compared to WT mice in the choroid plexus (Fig. 5g, h), one of the interfaces between the blood and the brain where early iron deposition occurs in cpKO mice[20]. CP treatment partially but significantly limited this accumulation (Fig. 5g, h), demonstrating that administered CP was effective in controlling iron accumulation in the choroid plexus of cpKO mice. Concomitant with iron accumulation in the choroid plexus, substantial cerebellar neurodegeneration has been reported in cpKO mice[20]. Indeed, significant loss of Purkinje neurons was observed in 10 months old cpKO mice compared to WT mice, and treatment with kCP effectively prevented neuronal loss (Fig. 5i, j). Intracellular deposits of lipofuscin, a well-known sign of neuronal degeneration[31], may foster cell body autofluorescence. Consistent with neurodegeneration, deposits of autofluorescence were observed in the cytoplasm of Purkinje neurons of cpKO mice (Fig. 5k, l), with a significant percentage of Purkinje cells of cpKO mice being positive ( + 58% *vs.* WT) and a prevention of this sign

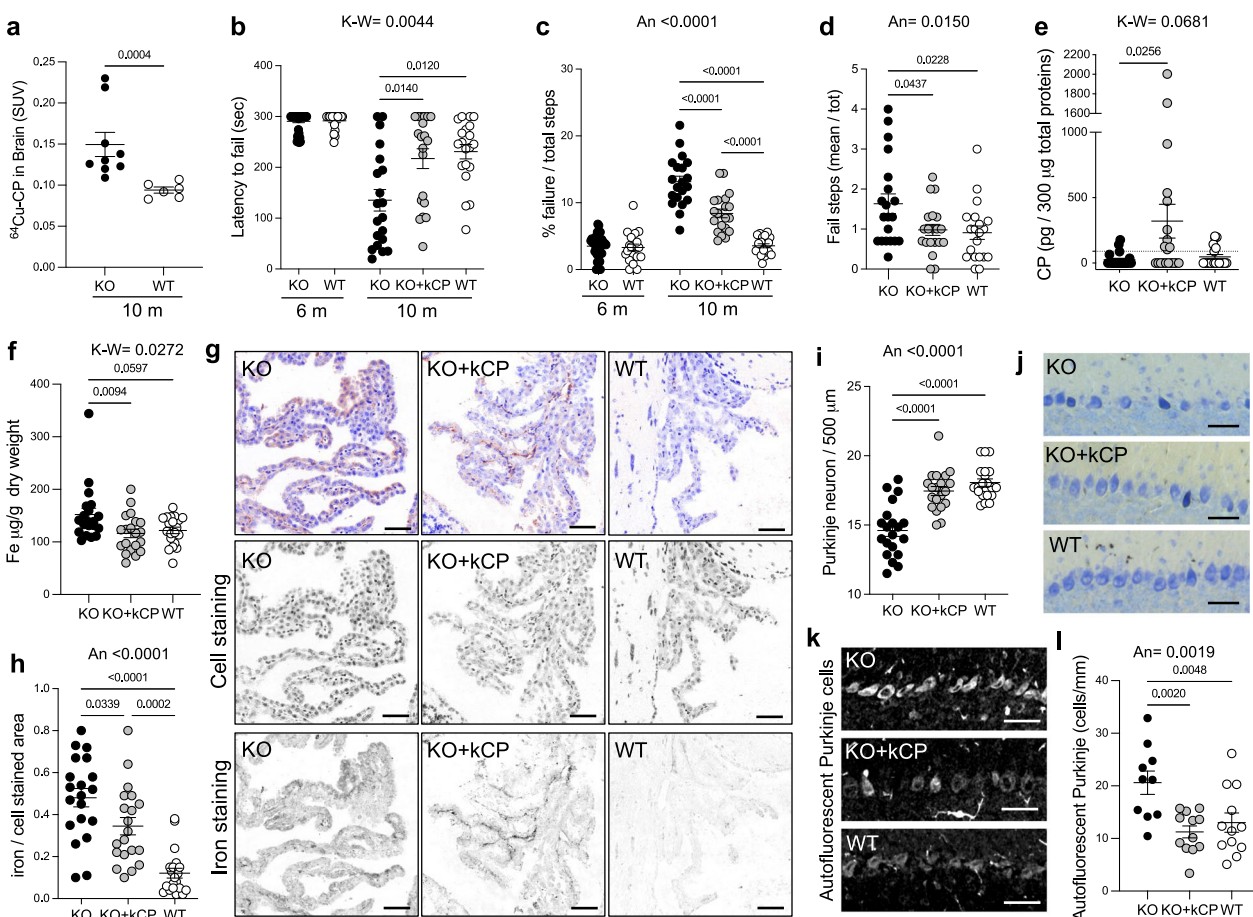

**Fig. 5 Administered kCP enters in the brain of cpKO mice inducing improvement in motor coordination, reduction of iron accumulation and prevention of cerebellar neurodegeneration. a** Ex vivo distribution of [$^{64}$Cu]-kCP measured in the brain of cpKO ($N = 9$) and control WT ($N = 6$) mice. Data are expressed as Standardized Uptake Value (SUV) and are the means ± SEM. **b–d** Motor coordination behavior analysis for **b** constant speed rotarod, **c** grid test, and **d** beam walking test in mice after 4 months of treatment with either kCP (KO+kCP) or saline (KO, WT). **e** ELISA evaluation of human CP in brain of treated mice. **f** Iron quantification by ICP-MS in whole brain homogenates. **g** Representative images of iron staining (brown) in choroid plexus where cells were counterstained in blue; gray-scale lower panels represent the originally acquired images for different wavelengths corresponding to blue and brown; magnification 20×, scalebar: 50 µm. **h** Quantification of iron deposition as ratio of iron/cells-stained areas. **i** Purkinje cells count on a linear distance of 500 µm in the cerebellum of treated mice. **j** Representative images of the staining of Purkinje cells; magnification 10×, scalebar: 50 µm. Data are presented as means ± SEM; each dot corresponds to one animal ($N = 20$ mice/group and $N = 40$ for cpKO at 6 months). Statistical $P$ values were evaluated by Mann-Whitney test in **a**, by one-way ANOVA (An) followed by Tukey's post-test analysis **c**, **d**, **h**, **i** or Kruskal-Wallis (K-W) test followed by either Dunn's **b** or uncorrected Dunn's **e**, **f** post-test analysis. **k** Representative confocal images of autofluorescent Purkinje cells. Scalebar: 10 µm. **l** Analysis of autofluorescent Purkinje cells linear density (KO $N = 10$, KO + CP $N = 12$, WT $N = 12$ mice/group). Data are presented as means ± SEM; each dot corresponds to one animal. Statistical P values were evaluated by one way ANOVA followed by Newman Keuls post-test.

of neuronal distress in kCP treated mice (Fig. 5k, l). As neurodegeneration is often tightly associated with neuroinflammation[32], we investigated the expression of glial markers GFAP (glial fibrillary acidic proteins) and IBA1 (ionized calcium-binding adaptor 1), markers of astrocytes and microglia, respectively[33], in the *arbor vitae* of the cerebellum (Fig. 6a). As previously reported[34], astrocytes were significantly less numerous in cpKO mice (-47% *vs.* WT), and treatment with kCP robustly mitigated this reduction (Fig. 6b). Also, GFAP expression levels were significantly lower in cpKO mice (-40% *vs.* WT), and treatment with CP prevented this effect (Fig. 6c). The density of microglia did not change in the 3 experimental groups (Fig. 6d) but overall expression of IBA1, a marker of microglia activation, increased significantly in the cpKO mice ( + 40% *vs.* WT) and treatment with kCP robustly prevented the increase in IBA1 expression (Fig. 6e). These data demonstrated that kCP treatment results in mitigation of neurodegeneration, astrocytic dysfunction, and microglial activation in the cpKO cerebellum, which can explain the observed amelioration of motor

coordination impairment in CP treated cpKO mice. Having observed the effects in the cerebellum, we examined the status of astrocytes (Fig. 6f), microglia (Fig. 6i) and neurons (Fig. 6l) in the hippocampus, a brain region not previously investigated in detail in cpKO mice. The analysis indicated a trend towards reduction of astrocytes in the CA1 *Stratum Radiatum* (SR) of cpKO mice in comparison to WT mice (-25% *vs.* WT) (Fig. 6g). The qualitative analysis (Fig. 6f) showed that in SR of cpKO mice astrocytes had thinner and shorter branches that in WT mice. Indeed, the expression of GFAP was significantly lower in the CA1 SR of cpKO mice (-61% *vs.* WT) and treatment with kCP robustly prevented this effect (Fig. 6h). These data demonstrated that astrocytes in CA1 SR of cpKO mice, though not significantly less numerous than in WT mice, demonstrated signs of cellular damage. Microglia status was evaluated in CA1 SR of mice using the anti-IBA1 antibody (Fig. 6i). The density of microglia cells was comparable in CA1 SR of the 3 experimental groups (Fig. 6j), similar to what observed in the cerebellum. However, in cpKO mice microglia had

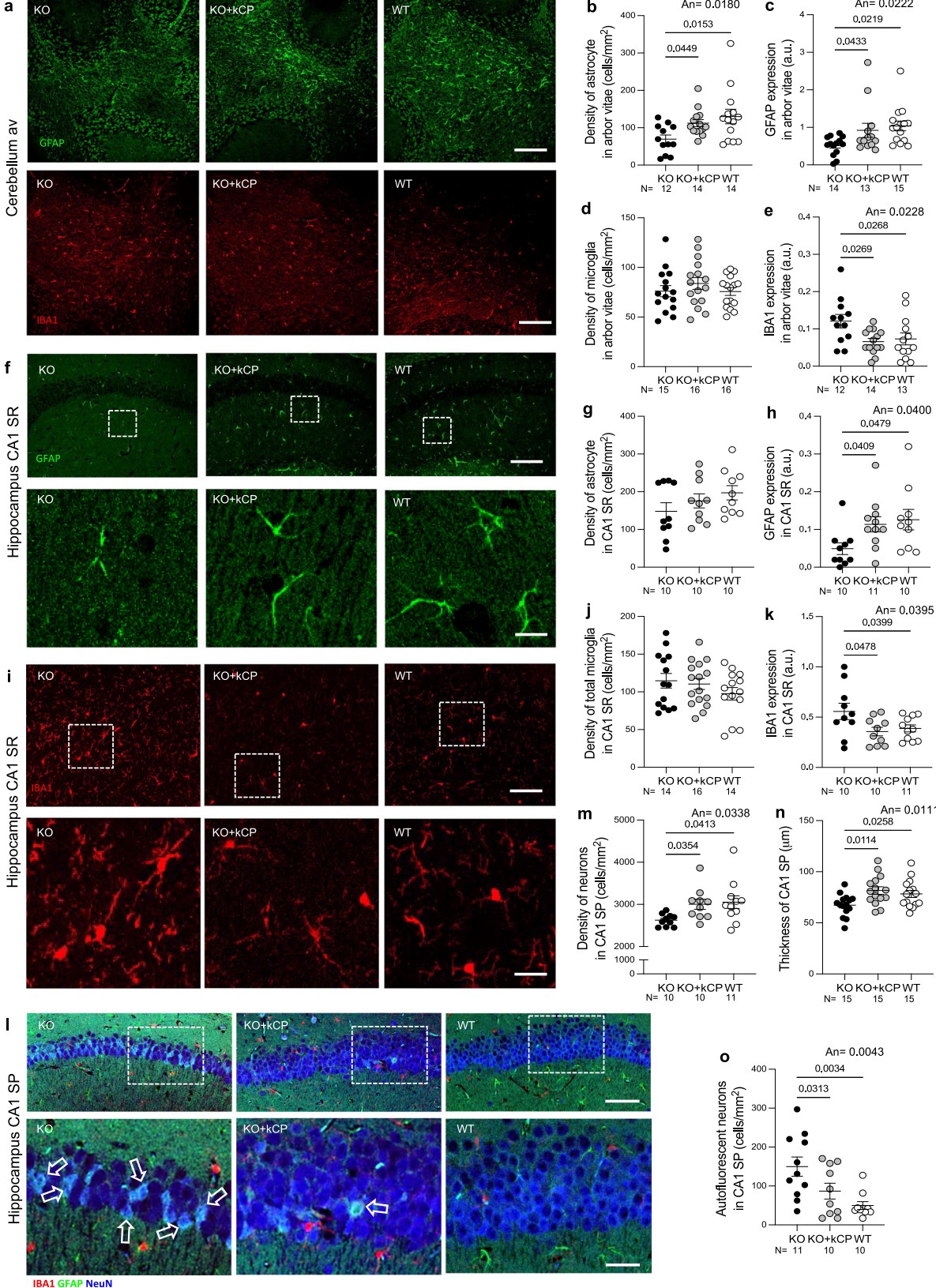

morphological features typical of activated cells, with bigger cell bodies and shorter and thicker branches than in WT animals (Fig. 6i). Indeed, we found a significant increase of IBA1 expression in CA1 SR of cpKO mice (+40% *vs*. WT), and kCP treatment prevented this effect (Fig. 6k). Finally, neurons were evaluated using the NeuN antibody in CA1 *Stratum Pyramidalis* (SP) (Fig. 6l). The density of neurons was significantly decreased in CA1 SP of cpKO mice (-14% *vs*. WT) and treatment with kCP significantly prevented neuronal loss (Fig. 6l arrows and 6m). In cpKO mice we found a significant thinning of CA1 SP (-14% *vs*.

**Fig. 6 kCP treatment prevents astrocytes damage and microglia activation in cerebellum and hippocampus, and mitigates reduction of CA1 pyramidal neurons in cpKO mice.** Immunohistochemical analysis of astrocytes, microglia, and neurons in the cerebellar arbor vitae and hippocampus of KO, KO+kCP and WT mice. **a** Immunofluorescent confocal images of astrocytes (green) and microglia (red) in the arbor vitae. Scalebars: 100 µm. **b** Analysis of astrocyte density, **c** GFAP staining, **d** microglia density, and **e** IBA-1 expression in the arbor vitae. **f** Immunohistochemical analysis of astrocytes in the Stratum Radiatum (SR) of CA1 hippocampus of sections of KO, KO+kCP and WT mice (upper panels, scalebar: 100 µm) and magnifications of the respective framed areas (lower panels, scalebar: 10 µm). **g** Analysis of astrocyte density and **h** analysis of GFAP in CA1 SR. **i** Immunofluorescent confocal images of IBA1 positive microglia in CA1 SR (upper panels, scalebar: 100 µm) and enlargements of the respective framed areas (lower panels, scalebar: 20 µm). **j** Analysis of microglia density and **k** analysis of IBA1 immunofluorescence in CA1 SR. **l** Confocal images of autofluorescent pyramidal neurons (white) in sections of CA1 Stratum Pyramidalis (SP) (upper panels, scalebar: 100 µm) and enlargements of the respective framed areas (lower panels, scalebar: 25 µm). Autofluorescent cells are indicated by open arrows. **m** Analysis of CA1 SP neurons density. **n** Analysis of CA1 SP thickness. **o** Analysis of autofluorescent pyramidal neurons density. Statistical *P* values were evaluated by one way ANOVA (An) followed by Newman Keuls post hoc test. All intensities are expressed in arbitrary units (a.u.) over a fixed threshold. Each dot corresponds to one animal, *N* is the number of mice analyzed for each group.

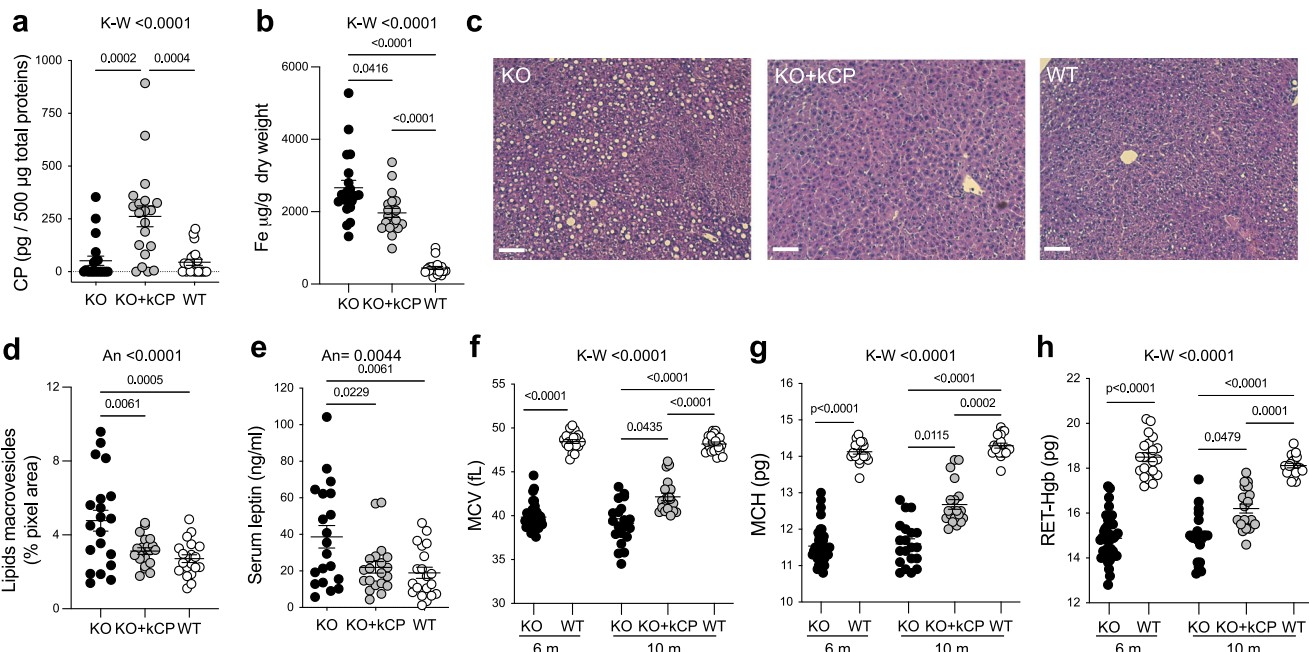

**Fig. 7 Systemic effects of kCP treatment on liver, adipose tissue and erythropoiesis of cpKO mice. a** ELISA evaluation of human CP in the liver of mice treated 4 months with either kCP (KO+kCP) or saline (KO, WT). **b** Iron quantification by ICP-MS in liver homogenates of treated mice. **c** Representative images of stained liver section in treated mice; magnification: 5×, scalebar: 100 µm. **d** Quantification of lipids macrovesicles in the liver of treated mice. **e** ELISA evaluation of leptin secreted by adipose tissue in the sera of mice. **f–h** Evaluation in the mice's peripheral blood of representative erythropoiesis parameters relevant for ACP, namely **f** mean corpuscular volume (MCV), **g** mean corpuscular hemoglobin (MCH) and **h** reticulocyte hemoglobin content (RET-Hgb). Data are presented as mean ± SEM; each dot corresponds to one animal (*N* = 20 mice/group; *N* = 40 for cpKO at 6 months). Statistical *p* values were evaluated by one-way ANOVA followed by Tukey's post-test **c**, **e** or Kruskal-Wallis test followed by Dunn's post-test analysis **a**, **b**, **f**, **g**, **h**.

WT), reflecting the decrease in CA1 neurons, and treatment with kCP prevented this effect (Fig. 6l, n). Deposits of autofluorescence were also observed in the cytoplasm of CA1 SP neurons of cpKO mice (Fig. 6l, arrows). The analysis of the autofluorescence signal demonstrated that a significant percentage of CA1 SP neurons of cpKO mice were autofluorescent ( + 115% *vs.* WT) and treatment with kCP strongly prevented this effect (Fig. 6o). Collectively, these results demonstrated that CP deficiency results in neuronal cell loss, astrocyte dysfunction and microglial activation in at least two brain areas (cerebellum and hippocampus) affected in aceruloplasminemia[16] and that kCP PRT can mitigate both neurodegeneration and neuroinflammation in these brain regions.

**kCP induces systemic phenotype amelioration in cpKO mice promoting reduction of iron accumulation and steatosis in liver, reduction of circulating leptin, and improvement of hematological parameters.** ACP is characterized by several systemic symptoms[16,17]. Since systemic dysregulation of lipid

metabolism in the liver and adipose tissue of cpKO mice and in ACP patients has been reported[16,21,35,36], we investigated the effects of kCP PRT in these tissue compartments. kCP administration (5 µg/g kCP administered to 6 months-old cpKO mice) for 4 months resulted in a significant CP accumulation in the liver of treated mice (Fig. 7a), reaching 1/7th to 1/75th of the levels of endogenous CP estimated in the liver of WT mice (ca. 11.3 pg/µg of liver extract[20]). Characteristically for the disease, significant iron accumulation was found in the liver of cpKO mice compared to WT mice and the treatment with kCP was efficacious in mitigating hepatic iron accumulation (Fig. 7b). Iron deposition in the liver of cpKO mice was concomitant with lipids accumulation highlighted by macro-vesicular hepatocytes[21]. A significant accumulation of lipids droplets was found in the liver of cpKO compared to WT mice (Fig. 7c), and a strong reduction of liver steatosis was evident in cpKO mice subjected to PRT (Fig. 7d). As prototypical readout of alterations in adipose tissue metabolism, levels of leptin (an adipokine released by body fat) were

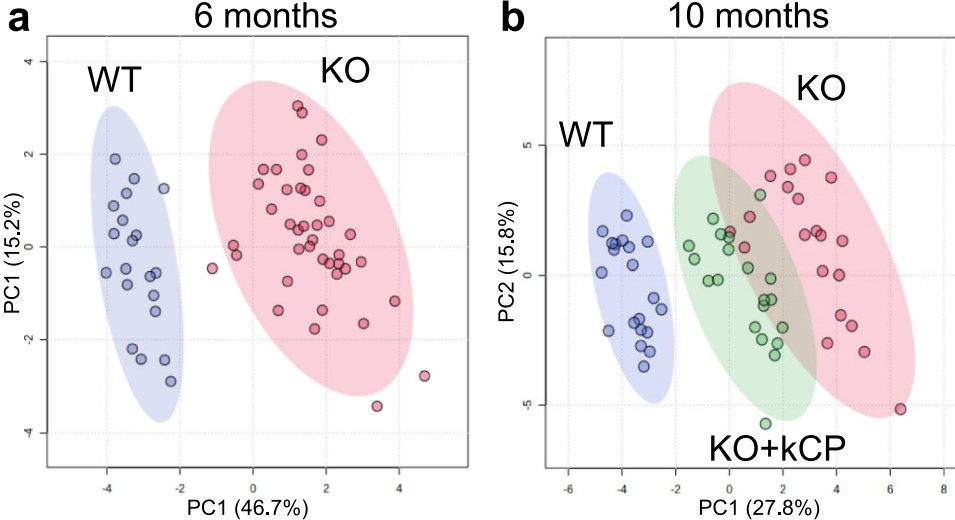

**Fig. 8 Unsupervised multivariate analysis of the parameters measured distinguishes cpKO from WT mice and indicates the therapeutic effect of purified kCP treatment.** Principal component analysis (PCA) performed **a** using 13 parameters measured in the cpKO and WT mice at 6 months of age, which included the 5 parameters shown in Figs. 4–7 plus animal sex and 7 additional hematological parameters, and **b** using 33 parameters, which included the 13 parameters shown in the figures plus animal sex and 19 additional features measured in the mice at 10 months of age after treatment with either purified human kCP (KO+kCP) or saline (KO and WT). The results of the immunofluorescence analysis in brain were excluded because data were available for only a subset of the mice. See Supplementary Tables 1 and 2, and Supplementary Data 5 for the values of the parameters included or excluded in the analysis. Each dot represents one animal; ellipses represent the 95% confidence interval area of each group.

investigated in the plasma of cpKO mice. Circulating leptin levels were higher in cpKO mice relative to WT counterparts and were significantly reduced upon kCP administration (Fig. 7e). These results provide further indication of the efficacy of PRT in preventing dysmetabolism in the liver and adipose tissue of cpKO mice. One of the hallmarks of the ACP, also recapitulated in the preclinical model, is an iron restricted erythropoiesis as consequence of poor iron availability[15–17,37,38]. Therefore, we investigated whether the prevention of iron accumulation in different organs, fostered by kCP administration in cpKO mice, also promoted the recovery of normal hematological parameters. The effect of kCP PRT in cpKO mice on three hematological parameters associated with iron availability, namely mean corpuscular volume (MCV), mean corpuscular hemoglobin (MCH) and hemoglobin in reticulocytes (RET-Hgb) were evaluated. Significantly lower MCV, MCH and RET-Hgb values than WT mice were observed in cpKO mice at both 6 and 10 months of age, confirming the impaired erythropoiesis (Fig. 7f, g and h)[37,38]. Treatment with kCP resulted in significant normalization of all three parameters (Fig. 7f, g and h), indicative of a substantial rescue of iron availability in cpKO mice.

**Unsupervised multivariate analysis discriminates cpKO from WT mice and indicates that PRT ameliorates the disease phenotype.** We applied an unsupervised multivariate analysis to evaluate the efficacy of kCP-PRT (5 μg/g kCP administered to 6 months-old cpKO mice for 4 months every 5 days) considering collectively all measured parameters, thus providing an aggregate view of the effect of PRT on the phenotype of cpKO mice (see Supplementary Data 5 for the values of the parameters included or excluded in the analysis). Principal component analysis (PCA) was performed using 13 parameters measured in the mice at 6 months of age (Fig. 8a), and 33 parameters measured in the mice after kCP PRT at 10 months of age (Fig. 8b) (see Supplementary Table 1 and Table2 for the averages of the additional parameters). These latter referred to hematological parameters, metals content in liver and brain, organs weight, liver inflammation and lipids dysmetabolism. The parameters referring to the

treatment, and not to its effect (e.g., CP expression level and immune response anti-CP), were excluded from the analysis. The PCA analysis clearly distinguished cpKO from WT mice (Fig. 8b). The cluster of cpKO mice, with its 95% confidence interval area, was evidently separated from that of WT mice, both at 6 and 10 months of age (Fig. 8a, b, red *vs.* blue dots). This result confirmed the presence of an overt pathological phenotype already in 6-month-old cpKO mice. At 10 months of age, the cluster of cpKO mice treated with kCP (green dots) clearly emerged from the cluster of untreated cpKO animals, shifting towards the WT mice cluster (Fig. 8b). This positioning, in between the two clusters, provides clear evidence for the therapeutic efficacy of PRT using kCP on the overall pathological phenotype of ACP.

**Discussion**
Optimization of use of plasma for therapeutics development is a major concern for both Society and Industry as human plasma is a precious, strategic resource resulting from donations. The global plasma fractionation Industry collects >50 million liters annually[3] and is worth >20B US$[39], with most plasma devoted to the production of immunoglobulins and albumin[39]. As immunoglobulin is the most demanded and valuable product, the Industry essentially collects plasma based on the anticipated demand for immunoglobulins[39]. Therefore, diverting plasma from the production of the most medically and commercially demanded products to produce new plasma-derived therapeutics decreases their availability and may result uneconomical, particularly for rare and ultra-rare disease indications. The use of unused plasma fractionation intermediates (an industrial waste during the manufacturing of medically valuable plasma-derived products) for the development of novel PRTs represents a key added value on several grounds. First, from an ethical perspective it optimizes the use of plasma, a scarce and precious resource resulting from donations, for therapeutics development. Second, from a medical perspective it enables the development of therapies for ultra-rare diseases where high cost of goods and small patient numbers may render their implementation uneconomical. Third, improving the economics of orphan drug development from plasma will

beneficially impact the cost of ultra-rare disease therapies with a benefit for patients and national health systems. Finally, reducing the industrial waste generated in the process of fractionating plasma, as such waste is re-introduced into the production cycle, will decrease the environmental impact of the Industry. To address these aspects, we systematically analyzed the proteome of the principal unused fractionation intermediates from an industrial plasma fractionation plant and found over 300 unique proteins represented in the unused intermediates aggregated proteome. Of the 244 proteins for which we found an association with human disease, 18 represented plasma products already on the market for different indications (e.g., coagulation and fibrinolysis proteins, immunoglobulin, albumin) and another 18 proteins were already in clinical trials, or the preclinical efficacy of a plasma-derived product was available in the literature. Within the latter subset, the most interesting and technically feasible from the perspective of developing a new therapy were considered to be ceruloplasmin (for aceruloplasminemia), haptoglobin and hemopexin (for hemoglobin toxicity relevant to sickle cell disease) and transferrin (for β-thalassemia), as preclinical efficacy studies were performed with proteins purified from plasma, representing effective comparators[20,23,40–42]. Interestingly, these proteins are all acute phase proteins associated with iron metabolism[43,44], which is dysfunctional in a plethora of diseases[45–47] and where these proteins may be therapeutic to address iron accumulation in tissues and detoxification of free hemoglobin[45]. Of these, we selected CP because of the ultra-rare nature of its primary deficiency[15], of its abundance in plasma[1] and of its stability in vitro and in vivo[20,24]. Finally, the use of plasma infusions to supplement iron chelation therapy in ACP provides a clinical rationale for CP replacement therapy[17,18]. CP is a multi-copper enzyme mainly expressed by the liver and secreted into the plasma at high concentrations (200 to 350 mg/L)[1,24]. CP plays several functions, and *inter alia* the ferroxidase activity is its major role which has clinical relevance[17,24,28]. As ferroxidase, CP is important in iron homeostasis promoting both the cellular iron efflux and iron incorporation onto transferrin for its transport into the plasma[24,48,49]. CP-deficiency leads to an intracellular iron accumulation, which promotes an increase of reactive oxygen species and consequent cell-toxicity, while at extracellular level it decreases the availability of oxidized iron fostering erythropoietic dysfunction and iron deficiency anemia[16,17]. Mutations in the CP gene are responsible for ACP, an ultra-rare monogenic recessive disease with adult onset. Clinically, ACP is characterized by iron deposition in organs (e.g., liver, pancreas, and retina), leading to liver dysfunction, diabetes, and vision reduction. These symptoms precede for several years the onset of severe neurological symptoms due to progressive neurodegeneration promoted by brain-iron accumulation[15–17]. No therapy is currently available capable of resolving neurological symptoms[16,17]. Since ACP is mainly a deficiency/dysfunction pathology of an extracellular protein, the opportunity to develop a PRT has stimulated our interest. In particular, the potential of PRT in reducing clinically relevant neurological symptoms of ACP, together with the systemic dysregulation of lipid metabolism, has been highlighted by studies that used human plasma-derived CP in a translationally relevant preclinical model of the disease[20,21]. This enabled us to compare the therapeutic properties of the CP purified from an unused intermediate of plasma fractionation with those of CP purified from whole plasma. The animal efficacy studies in cpKO mice reported here fully confirmed and substantially extended previous data on the translational pathognomonic phenotypes in cpKO mice and of the ability of a CP replacement therapy to mitigate them. The demonstration of the ability of $^{64}$Cu-labeled kCP, and of systemically administered kCP, to enter the brain in cpKO mice represents key CNS biodistribution evidence supporting the

restoration of the neurological and histopathological phenotypes by systemic PRT in cpKO mice. Although no direct data is available in ACP patients and despite the extreme rarity of the disease, some reports of the beneficial use of plasma infusions (Fresh Frozen Plasma, FFP, to provide the missing CP) in combination with iron chelation therapy are available[17–19] and is at least supportive of a PRT approach. FFP treatment in this context is generally provided as a weekly 500 ml infusion (representing ca. 2.5 mg/kg/week), which is broadly comparable to the dose we and others have used in mouse models (5 mg/kg every 5 days[20,26]). A clinically relevant aspect for PRT is the possible development of neutralizing anti-drug antibodies. The proportion of treated patients developing such responses is variable depending on the indication and the drug (e.g. see[50]). This can be at least partially addressed with immunotolerance induction approaches[51] and in some instances no antibodies against the therapeutic protein are detected (see e.g. plasminogen replacement therapy in PLG deficiency type I[52]). In the case of our study, we have addressed this experimentally in the mouse model and found that human CP does not stimulate the production of neutralizing anti-drug antibodies in the mouse (at least during the course of our experiments, 4 months), although this aspect will ultimately require testing in the clinic. Another therapeutically relevant aspect is the access of systemically delivered CP to the CNS. This may be possible through some yet unknown specific form of active transport across the CNS barrier systems (blood-brain and blood-cerebrospinal fluid). However, there is robust and increasing evidence that the blood brain barrier is progressively compromised in several neurodegenerative diseases[53], thus providing a likely rationale for CP PRT reaching the CNS under these conditions. Coherently, our studies with radiolabelled kCP in mice demonstrated higher brain levels of kCP protein in cpKO mice relative to WT mice, confirming previously reported immunoblotting data[20,26] and indicative of a brain barriers impairment in these mice. The higher kCP accumulation in the brain of cpKO mice might be due to a blood-cerebrospinal fluid-barrier leakage promoted by iron accumulation in choroid plexus epithelial cells we disclosed as early iron deposition event in CNS (re.[20], and this work). kCP accumulation in brain supports the feasibility of a systemic CP PRT therapy to address ACP neurological symptoms. Indeed, when dosing kCP in cpKO mice we observed a reduction of brain iron accumulation, amelioration of motor coordination and robust prevention of Purkinje neuronal loss, confirming previously reported data[20]. Importantly, for the first time, to the best of our knowledge, we observed hippocampal neurodegeneration in cpKO mice and demonstrated a robust neuroprotective role for kCP PRT in this brain region as well. The association between neurodegeneration and neuroinflammation is well established[54], which prompted us to investigate the co-existence of astrocytic and microglial dysfunction in cpKO mice. Astrocytes play a major role in iron handling within the brain, indeed, one of the hallmarks of the brain damage in ACP is the presence of dysfunctional/damaged astrocytes which show large intracellular iron deposition due to the lack of the expression of a glycosylphosphatidylinositol-anchored membrane isoform of CP[34,49,55]. However, the physiological role for this isoform in vivo remains unclear, and the ability of CP PRT to mitigate phenotypes in CP knock-out mice (ref.[20] and this manuscript) indicates that its lack in the CNS (as in the CP knock-out mouse model employed here) can be complemented by systemically delivered (or secreted) CP. We extended our finding on the rescue of the brain damage by observing that kCP treatment results in prevention of cerebellar astrocyte dysfunction, rescuing cell density likely by limiting the detrimental GFAP oxidative stress modifications described in the astrocytes of ACP patients[56,57]. Interestingly, we observed microglial activation in the brain of cpKO

mice, a previously undescribed component of the neuropathology in these mice which was also prevented by kCP PRT. To our knowledge, this demonstrated for the first time a beneficial impact on neuroinflammatory mechanisms in ACP. Neuroinflammation deserves to be carefully investigated since it might have a fundamental role in the onset and progression of the disease. In fact, signs of tissue inflammation in the absence of CP have been reported in the liver and adipose tissue of cpKO mice[21]. Therefore, the improvement of motor coordination promoted by kCP treatment in this model might be the effect of a limitation of neuronal and astrocytic degeneration and a mitigation of neuroinflammation. Previous studies on cpKO mice have reported loss of dopaminergic, retinal and cerebellar neurons[25], but the hippocampus had not been investigated in detail. In cpKO mice, deficiency in hippocampal neuronal functionality has been reported and linked to anxiety phenotypes[58], while in an astrocyte-conditional cpKO model reduced neurogenesis and memory defect implicated a hippocampal component[59]. Therefore, our findings of hippocampal neurodegeneration, associated to astrocytes dysfunction and presence of activated microglia shed new light on pathological mechanisms in this preclinical model of ACP. Additionally, the observation that kCP treatment results in prevention of hippocampal neuron degeneration, astrocyte dysfunction and microglia activation suggest that CP replacement therapy may also affect memory and cognitive symptoms, which are reported in ACP patients[16,17] and which deserve further investigation. The question as to the cause-effect relationship between the observed neurodegeneration and neuroinflammation in mice lacking CP, and to the potential for anti-inflammatory approaches in complementing iron chelation therapies currently administered to ACP patients, also provides ground for further studies. Iron deposition and lowered CP levels are features of other neurodegenerative conditions[60,61]. Indeed, low ceruloplasmin levels and brain iron accumulation are associated with both common and rare neurodegenerative diseases, such as Parkinson's disease, Alzheimer's disease and Wilson disease[16,55,60,62–65], and CP replacement therapy might represent as a potential adjuvant therapy in at least some of these conditions and in a relevant subsets of patients. At the systemic level, 4 months of treatment allowed to also detect circulating CP (differently from our previous study in which mice had been treated for 2 months), suggesting a small but constant protein accumulation in the blood that could reach para-physiological concentrations in prolonged treatments. The partial reconstitution of CP levels was proportionally higher in brain and liver than plasma, suggesting that the majority of administered CP was sequestered in the organs where iron deposition occurs. The mobilization of iron from brain and liver, promoted by kCP replacement therapy made iron available for the partial rescue of hematological parameters, demonstrating for the first time (to our knowledge) the efficacy of this approach also for the iron restricted erythropoiesis. From the treatment point of view this is extremely relevant because it overcomes the limitation of the iron-chelation therapy that is effective in reducing systemic iron accumulation but is not effective on brain iron deposition and it must be often discontinued due to the aggravation of functional iron deficiency anemia[55]. kCP replacement therapy exerted also additional beneficial systemic effects in cpKO treated mice, including prevention of iron accumulation and steatosis in liver, and the reduction of leptin circulating levels thus confirming previous indications on its capability to correct lipid dysregulation in ACP[21].

The studies presented here demonstrate the feasibility of the proposed approach. Limitations of the approach, include the limited sensitivity of the LC-MS methodology employed here that may exclude potentially interesting proteins from being detected and included in the analysis. Clearly, the presence of a protein in an unused fractionation intermediate does not guarantee equivalence of its analytical and functional properties to the same protein purified from whole plasma, which needs to be demonstrated both in vitro and in vivo. However, this is true of all plasma-derived proteins, where purification may potentially have an impact on the biochemical and biological properties of the protein of interest. Turning an exploratory prototype into a clinical candidate requires the production at scale according to the manufacturing and regulatory standards of the industry, as precedented for all plasma-derived therapeutics currently on the market. In the case of a CP replacement therapy for ACP, Orphan Drug status can be sought from Regulatory Agencies which facilitates therapeutic development in rare/orphan diseases. The extreme rarity of the disease requires the identification of an international network of clinical centers caring for ACP patients and the creation of a dedicated patient registry, currently not available, to support clinical development. Although the final objective must be full clinical development and registration to enable patients full access to the therapy, the absence of an effective treatment for the severe neurological late stage of the disease allows the consideration of a compassionate treatment approach for patients who progress to this stage even before the therapy is licensed. Finally, as for a majority of rare diseases a rapid and accurate diagnoses are of paramount importance to ensure ACP patients can receive a disease-modifying treatment as early as possible.

## Methods

**Plasma Fractionation waste fractions origin and processing**. Fraction I (FI), III (FIII) and IV$_{1-4}$ (FIV$_{1-4}$) were collected from plasma fractionation batches manufactured at Kedrion S.p.A. Bolognana plant according to[13], produced from plasma pools resulting from 5000-9000 individual donations collected as per applicable national and international legislation and industry standards[66]. FI is the filtration residue generated by cold addition to cryo poor plasma of ethanol 8%, celite and perlite and following alluvial filtration. FIII is the precipitate generated during IgG purification by resuspension of FII + III in acetate buffer, cold 17% ethanol addition, and alluvial filtration. FIV$_{1-4}$ is an intermediate generated within the albumin purification process by precipitation of resuspended FII + III with 40% ethanol and alluvial filtration (see Fig. 1a). Samples of the three solid intermediates were solubilized applying different conditions in which the impact of pH (5, 7, and 9) and buffer composition (water, 100 mM NaCl, 50 mM Na-citrate, Na-acetate, Tris, and Na-bicarbonate) were evaluated. The resulting suspensions were clarified by filtration (Cellulose filter sheets K900 – Pall) and characterized in terms of total protein content (Bradford assay) and electrophoretic pattern by SDS-PAGE (see below). The selection of the conditions for fractions resuspensions was based on feasibility of the trial (no clotting, high filterability in terms of time, pressure, volume) and characterization (higher total protein concentration and number of bands in SDS-PAGE).

**Shotgun Proteomics, sample preparation and analysis**. Proteins in the plasma fractions were quantified (Bradford method) and characterized by SDS-PAGE under reducing conditions (NuPAGE Novex 4–12% Bis-Tris acrylamide; Thermo Fisher Scientific) in MOPS buffer followed by Coomassie Blue staining. Plasma fractions were denatured in RIPA buffer and detergents were removed by filter-aided sample preparation (FASP). Proteins (200 μg) were diluted with 8 M urea and loaded on the filter-units (30-kDa cutoff, Microcon, Merck Group). Then 100 mM dithiothreitol were added to the samples in order to undergo

disulphide bonds reduction and after 30 min of incubation at room temperature filter-units centrifugation was performed at $13,800 \times g$ for 30 min. The thiol groups of the proteins were then alkylated using 100 mM iodoacetamide in 8 M urea for 30 min, after centrifugation the filter-units were washed with 8 M urea twice and successively with 50 mM ammonium bicarbonate twice. Proteins digestion was carried out on the filters at 37 °C overnight using Trypsin Gold-Mass Spec Grade (Promega) 1:50 in 50 mM ammonium bicarbonate. Peptides were collected by centrifugation, washed twice with 0.1% formic acid (FA). The tryptic peptides obtained from digestion of the plasma fractions were fractionated off-line using solid phase extraction cartridges HRP (Reversed Phase Polymeric) C-18 SOLA™ (by Thermo Fisher Scientific) in 8 different fractions using High-pH reversed-phase chromatography eluting peptides using buffers with 0.1% triethylamine and an increasing concentration of organic solvent (acetonitrile). Each plasma fraction was dried and reconstituted in FA (0.1%) to a final concentration of 1 µg/µl and analyzed in triplicate. Q Exactive™ HF-X hybrid quadrupole-Orbitrap™ mass spectrometer (ThermoFisher Scientific) was used to perform LC-MS/MS analysis. The peptide separation was carried out at 35 °C using a PepMap TM RSLC C18 column, $75 \times 150 \times 2$ mm, 100 Å (ThermoFisher Scientific) at a flow rate of 300 nl/min. The mobile phases A and B used were 0.1% FA in water and 0.1% FA in 80% acetonitrile, respectively. The gradient started with 5% of B, maintained constant for 5 min, then, the organic phase was increased up to 90% in 97 min and kept constant for 9 min and then returned to the initial conditions. Data dependent acquisition mode was used in which the most intense ions, a maximum of 12, from a full MS scan spectrum (200–2000 m/z) were selected for fragmentation.

**Bioinformatics analysis and protein prioritization**. Proteome Discover 2.5 (ThermoFisher Scientific) jointly with Sequest algorithm was used to identify the proteins. The reference database was *Homo sapiens* (Taxonomy ID: 9606), including 204961 proteins downloaded from UniProtKB (uniprot.org). Only proteins with FDR ≤ 0.01 were selected. The proteins identified in FI, FIII, FIV1-4 were characterized by an integrated databases analysis divided into four main steps: (a) the selection of identified proteins form the comparison with protein databases (The Human Protein Atlas, proteinatlas.org; and HUPO, hupo.org; Plasma Protein database, plasmaproteomedatabase.org) to select plasma proteins, (b) the association of proteins with relevant human diseases, (c) the existence of relevant animal models and other relevant tools (antibodies, assays, etc.) and (d) the evaluation of technical feasibility for the development of the identified targets into therapeutic approach. It was hence possible to confirm the presence of the proteins in the plasma and to gather information regarding Gene Ontology features (cellular component, biological process, molecular function), plasma concentration by immunoassay and/or mass spectrometry and the eventual existence of protein-specific antibodies. The obtained data for the fractions protein datasets underwent to a disease-association cross-analysis by assessing pathology and human genetics databases such as: DisGeNet (disgenet.org), MalaCards (malacards.org), GeneCards (genecards.org), HumanMine (humanmine.org), eDGAR (edgar.biocomp.unibo.it), DAVID (david.ncifcrf.gov). Additionally, for each disease more information was collected regarding its incidence, inheritance, age of onset, monogenic or multigenic evidence retrieved from OMIM (omim.org) and Orphanet (www.orpha.net). To have more reliable association data, the Gene-Disease Association (GDA) score was introduced. GDA ranges from 0 to 1 and it is computed according to a formula which considers the number and type of

sources (level of curation, organisms), and the number of publications supporting the association[22], GDA ≥ 0.7 is linked to a close and well-studied gene-disease relationship and was chosen as threshold value. Then animal models analysis was performed using International Mouse Phenotyping Consortium (mousephenotype.org) and Mouse Genome Database (informatics.jax.org). We also collected data about the phenodigm score which reflects the correlation between animal model and expected human disease phenotype[67]. The resulting subset of proteins was then compared with literature and patent databases (e.g. Pubmed; Espacenet, other published reports/patents for purification of protein from plasma) to evaluate technical feasibility of developing the identified targets through the various therapeutic modalities (hemoderivative, recombinant, gene therapy) and the presence of drug products addressing the medical need associated with the diseases, by consulting Drugbank (go.drugbank.com). All these steps resulted in protein prioritization that aims to identify plasma proteins from unused processing intermediates as a source for new protein replacement therapies. For the complete prioritization steps see Fig. 1c and Supplementary Data 1 and 2.

**CP Purification from Fraction IV₁₋₄**. Upon bioinformatic selection of CP as new target plasma derived therapeutic, the unused intermediates FI, FIII, and FIV₁₋₄ were characterized in terms of CP antigen level, ferroxidase and oxidase activity, purity and electrophoretic pattern to choose the most suitable starting material for its purification (see following paragraph). The CP purification was first based on an analysis of the relevant literature of the protocols for CP purification from whole human and animal plasma[27,68–73] and for commercial, research grade human CP available[20,26]. Additionally, some protocols for CP purification from plasma fractionation intermediates are present in the patent databases (e.g., US3003918A, 1961; JPS6187629A, 1986, RU2356560C1, 2007, RU2162338C1, 2001). CP was purified from solubilized FIV₁₋₄ by two ion exchange chromatographic steps followed by two viral clearance steps, formulation and lyophilisation, according to current applied procedures for plasma-derived therapeutics, resulting in a kCP concentrate powder, presented as 1 mg of CP antigen per vial, to be resuspended in 1 ml of solvent.

**kCP characterization**. Total protein content was quantified by Bradford method using bovine serum albumin (Pierce) as a standard, and human albumin 20% (Kedrion) as internal control. SDS-PAGE was performed using NuPAGE Novex 4-12% Bis-Tris acrylamide gel, MOPS SDS running buffer, NuPAGE LDS Sample buffer, and proteins were visualized by Simply Blue Safe Stain (ThermoFisher Scientific). CP purification intermediates were analyzed in semi-native denaturing (without boiling), denaturing (boiling samples at 90 °C for 10 min) and reducing conditions (by addition of NuPAGE antioxidant and Sample Reducing Agent, ThermoFisher Scientific). Cryocheck pooled normal plasma (Coachrom Diagnostica), together with commercially available research grade purified CP (ALX-200-089 ENZO LIFE SCIENCES) were used as reference. For Western blot (WB) proteins were transferred onto a nitrocellulose membrane with a Trans-Blot Turbo system (BioRad). Anti-CP antibody (ab19171, Abcam) and secondary antibody (Rabbit anti-goat Ig-HRP, Dako, Agilent P044901-2) incubations were carried out for 1 h at room temperature. Signals were revealed using chemiluminescence substrate (Clarity Western ECL Substrate – BioRad) on Chemidoc XRS+ (Bio-Rad Laboratories).

**Protein antigen detection**. For FI, FIII and FIV₁₋₄ CP antigen was measured by immunoturbidimetric assay (Optilite Caeruloplasmin

Kit; NK045OPT, Binding Site) using an Optilite apparatus (Binding Site), according to manufacturer's instructions. CP purification intermediates were analyzed by nephelometric assay using the N Antiserum to Human Ceruloplasmin (OUIE09, Siemens), the N Protein Standard SL (OQIM13, Siemens) as calibrator, and the N Protein Control SL/M (OQIO13, Siemens), onto a Siemens BN ProSpec system according to manufacturer's instructions. Analysis of contaminant protein antigen was performed by nephelometric assays with specific antisera (from Siemens) onto a Siemens BN ProSpec system according to manufacturer's instructions. The assayed antigens were Transthyretin (OUIF09 + Supplementary Reagent/Precipitation, OUMU15); Transferrin (OSAX09); α-2-Macroglobulin (OSAM09); Albumin (OSAL15); IgA (N Latex IgA kit, OQAI11); α-1-Antitripsin (OSAV09); Haptoglobin (OSAZ09).

**CP oxidase and ferroxidase activity.** CP oxidase activity on N,N-dimethyl-p-phenylenediamine substrate in waste fractions I, III, $IV_{1-4}$ and in purification process intermediates was determined by the EIACPLC—Ceruloplasmin Colorimetric Activity Kit (Invitrogen) at 562 nm using an ELX-808 reader (BioTek). Commercially available research grade CP products (ALX-200-089 ENZO LIFE SCIENCES and C4519 Sigma Aldrich) were used as controls. CP ferroxidase activity on ($Fe^{2+}$) substrate was determined in purified CP concentrate according to a modified Erel method[74] (see below). A calibration curve was generated by diluting CP standard included in the EIACPLC kit Invitrogen (range 148–29.5 mU/ml) with 45 mM Na-acetate, pH 5.8, and commercial CPs used as internal controls. The detection at 595 nm was carried out with a Synergy HTX reader (BioTek). Before analysis kCP lyophilized concentrate samples were dialyzed against saline solution, to avoid buffer interference as attested with citrate buffer.

**Mass spectrometry analysis of purified kCP.** Samples were digested in solution with trypsin at 37 °C overnight after being reduced and alkylated. They were then desalted and analyzed by UPLC-MS/MS using Q-Exactive HF-X (Thermo Fisher Scientific) as reported above. Amino acid sequence characterization by mass finger print peptide was also implemented. Sequence coverage mapping was performed using Biopharma Finder 2.0 software (Thermo Fisher Scientific). To identify eventual traces of other proteins, samples were also run with Proteome Discoverer 2.5.

**Mouse model and CP treatment.** cpKO mice (*mus musculus*, C57Bl/6 J)[25], and age and sex matched WT mice were used. The study was approved by the Institutional Animal Care and Use Committee and by the National Ministry of Health (n°77/2020-PR and no. 7/2022-PR). cpKO mice of 6 months of age ($n = 20$ each group, 10 males and 10 females) were treated for 4 months with either kCP purified from $FIV_{1-4}$ (5 µg/g in saline solution) administered intraperitoneal every 5 days or with saline solution as control. The WT mice ($n = 20$) were treated with saline solution with the same protocol. We have complied with all relevant ethical regulations for animal use.

**Behavioral tests.** The presence of neurological symptoms was evaluated by the assessment of motor coordination tests at the start and at the end of the kCP-treatment. For rotarod test the mice were placed on the apparatus with the rod rotating at 22 rpm and submitted to 5 trials with an interval of 30 min. The trials ended when mice fell down or after 5 min. The latency to fall was considered for each trial. The constant speed was defined as the average of the maximum speed reached by 5 WT and 5 cpKO mice that were placed on the rotarod rotating at 4 rpm and then the rotation speed was increased every 30 seconds by 4 rpm.

For the grid test each mouse was placed on a grid ($1 \times 1$ cm mesh) and left free to walk for 2 minutes under video camera recording. The total number of steps and the number of times that the animal's paws fall through the mesh of the grid were detected. The test was repeated 3 times and evaluated as number of failures normalized to the total number of steps. The beam walking test was performed on a 1-meter plywood rod with flat surface (10 mm), placed at a height of 50 cm. Three separated trials were performed, with 15 min of interval, and the number of failures of hind limbs over the path was recorded.

**Ex-vivo analysis of $^{64}$Cu-labeled kCP accumulation in blood and brain of cpKO and WT mice.** Human kCP purified from $FIV_{1-4}$ was radiolabelled with $^{64}$Cu by isotope exchange[75]. In brief, 1 mg of lyophilized kCP was resuspended in 1 ml ultra-pure water and buffer exchanged using Amicon Ultra-0.5 Centrifugal Filter Unit (UCF 505096, Merck) with cut-off 50 kDa. kCP protein (132 kDa) was concentrated in the filter and buffer replaced five times by centrifugation for 10 min, at 4 °C with sodium acetate buffer 0.2 M, pH 5.6; the final resuspension was to 1 ml. About 200 µl of buffered acetate kCP were then labeled with 10 mCi $^{64}$CuCl$_2$ (Acom), adding about 60 µl of ascorbic acid as stabilizer agent and sodium acetate buffer, under helium flux by spontaneous isotope exchange, incubating the mixture for 90 min at room temperature. Quality control was performed with HPLC and radio-TLC (thin layer chromatography), after adding the chelating agent diethylenetriaminepentaacetic acid (DTPA), for binding free radioactive copper present in the reaction solution. For radio-TLC, 5 µl of sample were seeded onto TLC plates (Agilent) run in physiological solution. For HPLC, a XBridge Protein BEH SEC (Waters) size exclusion chromatography column was used. As mobile phase 100 mM phosphate buffer pH 6.8 at 0.86 ml/min flux was used. The yield of the radiochemical reaction was about 20%, which in turn means that about 2 mCi of $^{64}$Cu were bound to kCP. Radiolabelled kCP was purified from free copper by using Amicon Ultra-0.5 Centrifugal Filter Unit (UCF 505096, Merck), cut-off 50 kDa (15 min, 14000 g at 4 °C), performing 3 time the buffer exchange with PBS and ascorbic acid; protein was finally resuspended at 1 mg/ml. The radiochemical purity of $^{64}$Cu-kCP was greater than 97% or equal to 100% as assessed by HPLC or radio-TLC, respectively, indicating that the labeling process did not lead to protein degradation.

For ex vivo analysis of CP accumulation in blood and brain, $5.12 \pm 0.25$ MBq of [$^{64}$Cu]-radiolabelled kCP were injected in the caudal vein of both cpKO ($n = 9$; 5 females and 4 males) and WT mice ($n = 6$; 3 females and 3 males) at 10 months of age. After 21 h from [$^{64}$Cu]-kCP administration the animals were anaesthetized, and blood was withdrawn by cardiac puncture. A portion of whole blood was transferred directly for the counting, while the remaining portion was centrifuged to obtain plasma. Thereafter, the animals were sacrificed by cerebral dislocation for brain sampling. Radioactivity in 100 µl of blood, 100 µl of isolated plasma and in whole brain was evaluated using gamma-counter (LKB Compugamma CS1282, Wallac). The data were expressed as SUV (Standardized Uptake Value), i.e., the concentration of radioactivity corrected for the injected dose and for the weight of the animal.

**CP evaluation by ELISA.** The level of human CP was evaluated by ELISA (Human Ceruloplasmin ELISA Kit, NBP2-60616 Novus Biologicals) in 10 µl of plasma and in 300 µg and 500 µg of brain and liver protein extracts, respectively. Absorbance was measured at 450 nm with microplate reader (iMark, BioRad).

**Evaluation of ferroxidase activity**. Ferroxidase activity was evaluated according to a modified Erel assay[74]. Three µl of plasma were added to 750 µl of acetate buffer solution (45 mM Na-acetate, pH 5.8), then 150 µl of substrate solution (367 µM ammonium iron(II) sulfate hexahydrate, 130 mM thiourea) were added and incubated 1 h at 37 °C. The colorimetric reaction was activated by adding 60 µl of 18 mM Ferene-S solution and absorbance measured in triplicate at 595 nm on a microplate reader (iMark, BioRad). As positive control 3 µl of 25 mM EDTA solution were used instead of plasma sample, while for negative/background control only reagents without sample were used. The activity was reported as percentages of the average of the activity evaluated in the plasma of WT mice.

**Detection of anti-CP antibodies in the plasma and analysis of their neutralizing effect on CP-ferroxidase activity**. The occurrence of anti-CP antibodies in the plasma of kCP-treated mice was evaluated by ELISA performed on purified human kCP. IgG fraction from mouse plasma was purified with rProtein-G-agarose beads (Invitrogen) as reported in[20] from 5 pools of 4 randomly selected sera from the same experimental group of animals (750 µg total proteins). The retention of the ability to bind human kCP by the purified IgG was tested by ELISA, using 150 ng of purified IgG from each pool of mouse plasma against 50 ng of purified human kCP. Neutralizing effect of IgG purified from mice plasma on the CP ferroxidase activity was analyzed by modified Erel assay. Purified kCP (600 ng) was incubated with 3 µg of purified IgG from each pool of mouse plasma, and ferroxidase assay was performed. kCP alone, heat-inactivated kCP (5 min at 100 °C), kCP incubated with 1 mM NaN$_3$ as selective inhibitor[30] or with 3 µg of a mixture (1/1/1 vol) of anti-CP commercial antibodies (Abcam: ab8813, ab48614, ab19171; Santa-Cruz: sc21242, sc21240) were used as controls. Oxidized iron was reported considering the absorbance of the total amount of ferrous iron present in the assay (3.2 µg) and was compared to the activity of purified kCP (1 µg).

**Analysis of iron by inductively coupled plasma-mass spectrometry (ICP-MS)**. Concentrations of metal ions was evaluated in lyophilized specimens of brain and liver of mice, after incubation (1 h at 70 °C) in a mixture of 65% nitric acid and 30% hydrogen peroxide, by ICP-MS using an ELAN DRC II instrument (PerkinElmer Sciex) as described in[20].

**Iron histochemistry**. Modified Perl's staining was performed on 14 µm sagittal sections of paraformaldehyde fixed brain as described in[20]. Staining at different emission wavelength was evaluated by the Nuance® FX multiplex biomarker imaging system (PerkinElmer) in the choroid plexus of the mice. A semi-quantitative analysis of iron deposition normalized for cell staining was performed using ImageJ software (National Institute of Health, http://rsb.info.nih.gov/ij) as in ref. [20].

**Purkinje cells count**. A total of 14 µm sections of paraformaldehyde fixed brain were stained with toluidine blue and analyzed with a Zeiss AxioImager M2m microscope. Images were acquired with AxioCam MRc5 (Zeiss) and analyzed with ImageJ software for automated count of Purkinje body cell over a linear distance of 500 µm in the Purkinje cell layer of the cerebellum. Five different linear distance were analyzed for each mouse.

**Fluorescence Immunohistochemistry**. Immunohistochemistry was performed on paraffin-embedded mouse brain sagittal sections (5 µm) after deparaffinization and antigen retrieval. Triple immunostaining for NeuN+, GFAP+, IBA1+ cells was performed using a mouse anti-NeuN antibody for neurons (1:400; MAB377, Millipore), a rabbit anti-IBA1 antibody for total microglia (1:300; 016-20,001, WAKO) and a mouse anti-GFAP antibody conjugated with Alexa Fluor 488 (1:500; MAB3402X, Millipore) for astrocytes[76]. Alexa Fluor 635 anti-rabbit (1:600; Alexa Fluor 555 anti mouse antibody (1:600; A31577 and A31570, Thermo Fisher Scientific) were used as secondary antibodies.

**Confocal microscopy and quantitative image analysis**. Confocal scans were acquired on a LEICA TCS SP7 (Leica Microsystems CMS GmbH) equipped with 20X objective (z step 0.6 µm). Analyses were performed stacking 10 consecutive z scans with ImageJ software. Immunofluorescence of GFAP, IBA1 were detected setting a fix threshold level with ImageJ threshold tool and the positive pixels above threshold were quantified and expressed as percent positive pixels/total pixels[76]. The number of astrocytes, total microglia, NeuN +, and autofluorescent pyramidal neurons were evaluated using the multi-point tool and expressed as cells/mm$^2$. Autofluorescent Purkinje cells were expressed as cells/mm. The thickness of CA1 was evaluated using the line tool and expressed in µm.

**Liver histological analysis**. Automated hematoxylin-eosin staining was performed on 3 µm sections of paraformaldehyde fixed and paraffin embedded liver using the Leica BOND RX instrumentation at the Animal Histopathology facility, HSR. Samples were analyzed with Zeiss AxioImager microscope. Lipid microvesicles were quantified on histological images with ImageJ software. Four images for each mouse were segmented by a threshold filtration to define white lipid droplets staining, and then quantified as percentage of the pixel area covered by lipid droplets on the total of pixel area.

**Leptin evaluation**. Leptin level was quantified using ELISA kits (MOB00, R&D Systems) in mouse plasma samples (diluted 1:40) and plates were read in a BioRad iMark spectrophotometer. The optical density was determined by subtracting value obtained at 570 nm of wavelength from value obtained at 450 nm according to manufacturer's instructions.

**Hematological parameters evaluation**. Blood collected from mice at 6 and 10 months of age was analyzed for hematological parameters at the San Raffaele Mouse Clinic-Animal Biochemistry facility using ILab Aries (Instrumentation Laboratory) instrumentation for biochemistry analyses, and Idexx Procyte analyzers for hematology.

**Statistics and reproducibility**. Statistical analyses were performed with Prism V9.5 software (GraphPad Inc.), a two-tailed $p$ value < 0.05 was considered significant comparing means ± standard error of the mean. Data from two groups were evaluated by unpaired Student's t-test, if they passed the normality test (Kolmogorov-Smirnov test) or were evaluated by Mann Whitney test. Differences between three groups were assessed by one-way Analysis of Variance (ANOVA) if data passed the normality test or by Kruskal-Wallis test if they don't. Post hoc test analysis was performed to compare all pairs of groups, namely Tukey's test or Newman-Keuls' test for ANOVA and Dunn's or uncorrected Dunn's test for Kruskal-Wallis. The sample size ($n = 20$ animals per group) was established using the G-Power v3.1.9.4 software (Heinrich-Heine-Universität Düsseldorf), applying a one-way ANOVA test for the comparison of means between 3 groups with alpha error of 0.05 and power of 0.8; effect size (Cohen's effect size) of 0.45 which by convention is a large effect size. The resulting number was $n = 17$ animals per group, however, we increased the number by 20% to compensate for any/possible animal

deaths which, from our previous experience[20] can occur with the animals aging up to 10 months. Multidimensional reduction analysis was done using unsupervised principal component analysis (PCA). Data were normalized by auto-scaling transformation, and the analysis was performed using MetaboAnalyst 5.0 online package (www.metaboanalyst.ca/home.xhtml). Upon PCA, the experimental groups of animals were highlighted as 95% confidence interval area automatedly defined by the software.

**Reporting summary**. Further information on research design is available in the Nature Portfolio Reporting Summary linked to this article.

## Data availability

The datasets of the study are available at "San Raffaele Open Research Data Repository" (ORDR) (https://ordr.hsr.it/research-data/); https://doi.org/10.17632/r2p5n38mdz.1. The mass spectrometry proteomics data have been deposited to the ProteomeXchangeConsortium via the PRIDE partner repository with the dataset identifier PXD046234.

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

## Acknowledgements

The funding supporting this work are: Kedrion S.p.A., SR-03/19 (M.A.); Italian Ministry of Health, grant Ricerca Finalizzata 2018, RF 2018-12366471 (M.A.); Ministero dell'Istruzione, dell'Università e della Ricerca, grant [giovanniniricaten2023] (M.G.G.): Fondazione Cassa di Risparmio di Firenze, grant [PROPREBIOAD] (M.G.G.). D.L. was recipient of a fellowship from Fondazione U. Veronesi (Post-doctoral Fellowships 2023) and his current position is supported by #NEXTGENERATIONEU (NGEU) and funded by the Ministry of University and Research (MUR), National Recovery and Resilience Plan (NRRP), project MNESYS (PE0000006)(DN. 1553 11.10.2022). We thank the Advanced Light and Electron Microscopy BioImaging Center, Animal Histopathology and Biochemistry facilities at the IRCCS Ospedale San Raffaele for assistance. We thank the Institute of Applied Physics "N. Carrara", National Research Council (IFAC-CNR), Italy for use of confocal laser microscopy equipment. Dr. Alessandro Gringeri current address is Fondazione Charta, Center for Health Associated Research and Technology Assessment, Milan, Italy.

## Author contributions

Conceptualization: M.A., A.Ca. Methodology: A.Z., G.D.P., I.N., C.S., M.G.G., V.C., R.M.M., P.D. Investigation: A.Z., S.R., B.F., A.Con, E.G., I.N., F.M., L.N., A.P., P.G., G.Z., G.Magh, G.T., D.L., G.Magn, L.S., L.T., C.T., C.M., A.Col, S.B., P.R. Funding acquisition: M.A., A.Ca, M.G.G. Supervision: M.A., A.Ca. Writing, original draft: M.A., A.Ca, A.Z., I.N. Writing, review & editing: M.A., A.ca, A.Z., I.N., A.G., M.G.G., D.L., C.S., V.C. All the authors discussed the results and commented on the manuscript at all stages.

## Competing interests

Authors declare that they have no competing interests. M.A. and A.Ca disclose that the study was partially supported by Kedrion S.p.A. within the frame of a scientific collaboration. A.Ca, I.N., F.M., G.Z., C.S., L.N., A.P., P.G., G.Magh, G.T. are employees of Kedrion S.p.A.
