## [Peer Review File · Communications Biology]

Reviewers' comments:

Reviewer #1 (Remarks to the Author):

The manuscript by Zanardi et al have identified 308 unique human plasma proteins from discarded fractions of an industrial plasma fractionation plant. 68 of these unique proteins were associated with 115 disorders, they then went on to rank proteins based on a number of parameters, including, association with ultra-rare disease which can be treated with a therapy including plasma. They chose Ceruplasmin deficiency which is associated with a genetic mutation in the CP gene and an overall reduction in CP protein. CP protein was then purified and characterised from discarded plasma fractions using a number of assays to ensure CP integrity, purity, and activity. They administered this purified active CP into mice and demonstrated that the protein does not cause acute toxicity, following on from this, they injected CP (which now they refer to as kCP) over 4 months to WT and cpKO mice to demonstrate PRT therapy. kCP treatment rescued protein level and ferroxidase activity in the plasma of cpKO mice, kCP was able to enter mice brain and reduce iron accumulation and prevent neurodegeneration. kCP treatment was also able to reduce steatosis in liver and improve haematological parameters. Overall, this manuscript is a large body of work demonstrating that CP replacement therapy for ACP is effective in improving disease phenotype.

Major comments to be addressed

1. Is plasma therapy for ACP in the clinics similar to this study's treatment condition, can you please compare and include this in your discussion? It's also worth discussing administration, dosage and eliciting an immune response in humans, and its limitation/benefits in human studies and comparing this to current PRT studies in the clinics.
2. Did you check for iron accumulation in the retina and pancreases of mice?
3. What would happen if you were to inject your CP protein to healthy mice (WT), you have acute toxicity data, however, your experiment is over 4 months, not 5 days (as shown for the acute study). Can you please discuss this or do you have any data on healthy mice treated with CP over 4 months?
4. For the proteomics study on the discarded plasma fractions, the plasma used was a pool of plasma from many individuals, how many? Further, for protein identification, how many peptide counts were included for a protein to be "uniquely identified". Was any PTMs included in the analysis parameter? The analysis is very vague, please include more detail: Line 507 – what is this semi-automated bioinformatics workflow, can you provide brief description? Further, proteomics sample prep is not clear – the samples were run through filtration unit prior or after reduction and alkylation? Please make it clear, also what speed in the centrifuge was the filtration unit spun for? Why were filtration units used, to remove detergents, if that is the case, then why was the washes with UREA introduced? How many fractions were generated from the solid-phase extraction cartridges, what was the condition? What is the intensity of these proteins identified from your proteomics data – how many peptides etc, please include this in your SM1.
5. What are these known proteases that cleave CP protein resulting in different band sizes on the western blot? Can you include this in your discussion? Further, why are those bands also present in the CP alone sample, it should not have any other contaminant proteins, please discuss.
6. For your Figures, some of them had detailed stats, while others were missing stats, see below for a good example from one your figures, please apply this to all. "Data are presented as mean \pm SEM; each dot corresponds to one animal (N= 20 and N= 40 for cpKO at 6 months). Statistical P values were evaluated by one-way ANOVA (An) followed by Tukey's post-test analysis (panels B,C,G,H) or Kruskal-Wallis (K-W) test followed by either Dunn's (panel A) or uncorrected Dunn's (panels D,E) post-test analysis."
7. Why weren't 33 parameters used to make Fig 8A, why were those parameters excluded, it is not clear, please discuss.

Minor comments to be addressed

1. Fig1b – you can see a dark faint line around the figure, please fix
2. Explain what GFAP and IBA1 is for non-expert reader?
3. The stats for ACP are in the discussion, I would prefer it in the introduction, can you can please

move this?

4. Why does lipid and iron accumulation correlate in the liver? Please discuss.
5. Line 427 – missing a word
6. Line 461 should be are
7. How were samples prepared for ICP-MS, a brief description is required?
8. Why does it jump from SM3 to SM6, why isn't it just SM4?
9. Line 90 needs to be re-written, what do you mean "associated with a known Homo sapiens gene"?
10. In Fig6, what does AN= MEANS?
11. Fig5, H and I are switched around. Fig 3, A2 and 3 are switched around. Please ensure the correct figure corresponds to the correct legend and also within text.
12. FigSM4.2 – why can't you detect the proteolytic versions of CP in the plasma of mouse?
13. FigSM5.1, there seems to be a slight decrease in body weight at 5ug/g, can you explain this,
14. Where is the raw proteomics data deposited?
15. It's not clear how long mice were treated with kCP, can you make it clear prior to every results section.
16. Line 196 – can you state this significant difference in number, this applies to most situation in the results section, where you have mentioned that there is a significant difference, it would be good to see what that number is and the associated p-value.

Reviewer #2 (Remarks to the Author):

Zanardi et al. describe the proteomic characterisation of unused plasma fractions, and the isolation and utilization of ceruloplasmin from these fractions. The proteomic analysis undertaken appears to be of high quality, however further description of the quantitative analyses carried out is needed.

Minor comments:

1. The authors describe the relative quantification of CP in the fraction FIV1-4 using mass spectrometry, however the relevant methods (pg. 19) do not describe how the quantitation was done. Table SM3 does not provide more context in this regard. Was the quantitation done through ProteomeDiscoverer using a label-free method? A more detailed description is needed in the methods to describe how the quantitation, and not just identification, of proteins in fraction FIV1-4 was achieved.
2. Relatedly, the authors note that in their proteomic analyses they estimate 98% purity of CP, as opposed to 80% purity by ELISA (pg. 6, line 138). I note that the authors use 30-kDa cutoff molecular weight filters in their sample preparation for mass spectrometry. This raises the question as to whether proteins smaller than 30 kDa are being lost in the sample processing, and whether this is skewing the relative abundance of CP in the sample? The authors should consider this in the interpretation of their results.
3. The authors mention that peptides were fractionated offline with SPE (pg. 18, line 492). More details are needed here. What specific cartridges were used (what chemistry), how was the fractionation done, and how many fractions were collected/analyzed?

Reviewer #3 (Remarks to the Author):

This manuscript (COMMSBIO-23-3004-T) is the second proof-of-concept study from Dr. Alessio's group testing whether ceruloplasmin (CP) isolated from human plasma can be used as a therapeutic agent

for patients with Aceruloplasminemia (ACP). ACP is a great candidate for PRT approach because it is a monogenic disease, and CP works extracellularly. In summary, these two studies are rather similar besides the source of CP with potentially a new extraction method and duration of treatment. However, this study can be a valuable addition to the field of rare diseases because they also found other diseases (interestingly all iron metabolism-related) that might be candidates for the unused plasma-driven therapeutics approach. I recommend this manuscript to be accepted for publication with some revision. Detailed critique below.

1) Please strengthen the immune response data

Many PRT or ERT (enzyme replacement therapy) induce immune response to foreign protein, therefore, immune response to kCP is one of the key pieces of data if this type of therapeutic approach goes to clinic. In this manuscript, the data is only available in the supplemental data section (SM 5.4) and there is barely any discussion in the manuscript.

2) What if the anti-CP antibody interferes with CP/FPN interaction?

CP is a ferroxidase, therefore testing the ferroxidase activity in the presence of anti-CP antibody is quite logical in Fig SM 5.4-C. However, CP as a ferroxidase is involved in iron efflux partnering with the sole iron efflux protein FPN. If this anti-CP antibody interferes with CP/FPN interaction, it will diminish cell's iron efflux capacity and cause ACP-like situation. This possibility needs to be addressed before we can deem this anti-CP antibody non-problematic.

3) ^{64}Cu -CP distribution experiment is quite important.

As mentioned above, ACP is probably one of the most logical targets for PRT because the secreted form of CP is abundant in plasma and produced by liver. However, CNS expresses GPI-anchored form of CP and isolated behind BBB. Although expected from the improvement of CNS phenotypes in their 2018 paper, it is quite cool to see that the radio-labeled CP was detected directly inside brain. In my opinion, this is a piece of very important data and yet hidden away in SM 5.3.

4) Discussion on possibility of secreted, BBB-crossed kCP to fill in the role of GPI-CP

As mentioned above, CNS expresses alternatively spliced form of CP, and it seems not clear why brain prefers anchored protein unlike plasma. It is quite interesting that human plasma-isolated secreted CP can alleviate neurological phenotypes in Cp null mice. A section in Discussion regarding this point would be beneficial to the field.

Reviewer #1 (Remarks to the Author):

The manuscript by Zanardi et al have identified 308 unique human plasma proteins from discarded fractions of an industrial plasma fractionation plant. 68 of these unique proteins were associated with 115 disorders, they then went on to rank proteins based on a number of parameters, including, association with ultra-rare disease which can be treated with a therapy including plasma. They chose Ceruplasmin deficiency which is associated with a genetic mutation in the CP gene and an overall reduction in CP protein. CP protein was then purified and characterised from discarded plasma fractions using a number of assays to ensure CP integrity, purity, and activity. They administered this purified active CP into mice and demonstrated that the protein does not cause acute toxicity, following on from this, they injected CP (which now they refer to as kCP) over 4 months to WT and cpKO mice to demonstrate PRT therapy. kCP treatment rescued protein level and ferroxidase activity in the plasma of cpKO mice, kCP was able to enter mice brain and reduce iron accumulation and prevent neurodegeneration. kCP treatment was also able to reduce steatosis in liver and improve haematological parameters. Overall, this manuscript is a large body of work demonstrating that CP replacement therapy for ACP is effective in improving disease phenotype.

Authors' remarks to the Reviewer:

We thank this reviewer for appreciating our manuscript and for providing constructive comments, which we believe we addressed through an in depth revision of the manuscript (modified/new text **highlighted in yellow**, with reference to relevant lines in the text).

Major comments to be addressed

1. Is plasma therapy for ACP in the clinics similar to this study's treatment condition, can you please compare and include this in your discussion? It's also worth discussing administration, dosage and eliciting an immune response in humans, and its limitation/benefits in human studies and comparing this to current PRT studies in the clinics.

Author's answer: Plasma therapy dosage in ACP is 500 ml/week, generally in combination with iron chelation (e.g. Poli et al., 2017; **38**:357-360. doi: 10.1007/s10072-016-2756-x; Tridimas et al., 2021, JIMD Rep. 2021 Jan; **57**: 23–28 doi: 10.1002/jmd2.12176). Although the original citations we cited in the manuscript (Myiajima et al., 1993) report CP plasma concentrations in normal individuals of 200-350 mg/L, more recent analyses involving large cohorts of individuals report a range of 200-500 mg/ml; e.g. see Yang et al. 2022 Front. Neurol. 13:1058642. doi: 10.3389/fneur.2022.1058642). Therefore, a 500mL plasma transfusion delivers ca. 175 mg CP (conservatively), once weekly (=700 mg/month). For an average adult patient of 70 kg, this corresponds to ca. 2,5 mg/kg CP per week. This dosing is broadly comparable to the 5 mg/kg every 5 days we employed in this mouse study. On a monthly basis, the dosing of 5 mg/kg every 5 days delivers 2100 mg per month for a 70 kg individual (3x the monthly dose achieved with FFP therapy in ACP patients). This compares well with current dosing of other protein replacement therapies for ultra-rare disease, for instance Rypplazim (plasminogen replacement therapy for plasminogen deficiency type I) is dosed at ca. 1000 mg/week (ca. 4000 mg/month), aimed at achieving 10-20% of normal plasminogen plasma levels (which is 100-200 mg/L; e.g. Shapiro et al. 2018, Blood 22;**131**:1301-1310. doi:

10.1182/blood-2017-09-806729). We have included a sentence in the discussion to cover this (lines 408-420).

As for immune responses in humans, stimulation of immunogenic responses during protein replacement therapy in human patients is known, though the proportion of treated patients developing such responses is variable depending on the indication and the drug (e.g. see De La Fuente et al., 2021 Int. J. Mol. Sci. 2021, 22(17), 9181; doi.org/10.3390/ijms22179181). Though antibodies against the therapeutic protein may be developed these are often non-neutralizing, and therefore do not represent a significant clinical issue. In some instances, no antibodies against the therapeutic protein are detected (see e.g. plasminogen replacement therapy in PLG deficiency type I; Shapiro et al., 2018; Blood **131**:1301-1310. doi: 10.1182/blood-2017-09-806729). In the case of our study, we have addressed this experimentally in the mouse model and found that human CP does not stimulate the production of neutralizing antibodies in the mouse (at least during the course of our experiments, 4 months). Investigations on the development of anti-drug antibodies is mandatory in clinical trials and will be investigated in Phase I in the case of CP replacement therapy for ACP. In any case, even in the presence of an immunogenic response (e.g. see in Hemophilia A) this can be at least partially addressed with immunotolerance induction approaches (e.g. increased dosing of the drug; Jardim et al. 2020, Res Pract Thromb Haemost **4**:752-760. doi: 10.1002/rth2.12335). We have included a sentence in the discussion to cover this (lines 410-420).

2. Did you check for iron accumulation in the retina and pancreases of mice?

Author's answer: Unfortunately we didn't because retinal iron deposition was reported in cpKO mice older than 16 months but not in 10-months old mice as used in our study (Patel et al. 2002; J Neurosci. **22**:6578-6586. doi: 10.1523/JNEUROSCI.22-15-06578.2002.). Retinal iron deposition was reported in younger mice (9 months) only in the case of animals double mutants for Cp and hephaestin (another ferroxidase; Hahn et al. 2004, Proc Natl Acad Sci U S A. **101**:13850-13855. doi: 10.1073/pnas.0405146101.), thus suggesting a mutual compensatory mechanism of the two molecules in the younger mice. In the case of pancreas iron deposition was not investigated because we were most interested in therapeutic effect of the PRT on the neurological symptoms, for which currently there is no therapy, and extended proof of concept for treatment efficacy only in some of the systemic symptoms. The interesting further developments suggested by this reviewer will be addressed in future dedicated studies.

3. What would happen if you were to inject your CP protein to healthy mice (WT), you have acute toxicity data, however, your experiment is over 4 months, not 5 days (as shown for the acute study). Can you please discuss this or do you have any data on healthy mice treated with CP over 4 months?

Author's answer: As the ultimate objective of a ceruloplasmin PRT is to treat aceruloplasminemia patients and not healthy individuals, we only included in our study a small (N=10) parallel satellite group of WT mice, which were treated for 4 months (from 6 to 10 months of age) with purified kCp with the same protocol used for the therapeutic treatment of cpKO mice. In this group of animals (5 males and 5 females), we evaluated only

selected parameters and behaviours, as well as gross signs of toxicity, namely changes in body weight and external appearance (e.g. fur appearance such as smooth/shiny/ruffled/bristly fur), the eyes (clear and shiny, or closed and not clean) and the general state of cleanliness as an index of normal grooming activities. Spontaneous and social behaviour were also observed (normal or reduced activity, isolation from the group) and the response to external stimuli (normal or limited depression or exaggeration of the response). None of these parameters resulted to be altered during the treatment with purified kCp in WT animals as also reported for KO animals. At the end of the treatment WT mice were evaluate for motor coordination behaviour tests (rotarod and beam walking assays) and animals euthanized for major organs (heart, liver, spleen, kidney) collection and evaluation (visual inspection and weight). No significant differences were found in motor coordination and organs appearance/weight of the WT kCp-treated mice in comparison with WT saline-treated animals. Nevertheless, a complete panel of GLP toxicology studies will be mandatory in future for the potential clinical development of the purified kCp. The results obtained with this satellite group of WT mice treated with purified kCp were introduced as new supplementary materials (SM5.2) in the revised version of the manuscript and reported/commented in the results section (lines 165-167).

4. For the proteomics study on the discarded plasma fractions, the plasma used was a pool of plasma from many individuals, how many? Further, for protein identification, how many peptide counts were included for a protein to be “uniquely identified”. Was any PTMs included in the analysis parameter?

Author’s answer: Plasma samples represent pools from 7000 individuals (on average), and we have added a sentence to specify this in the Materials and Methods (lines 514-515). For protein identification we considered at least one unique peptide. Furthermore, the coverage, the number of peptides, the number of unique peptides and the average of the abundance based on peak intensity are included in SM1 for each protein of every fraction, as suggested by the reviewer. In the analysis, only default PTMs (Oxidation / +15.995 Da (M) and Carbamidomethyl / +57.021 Da (C)) were included. Other PTMs were not investigated. Ceruloplasmin PTMs analysis would indeed be of interest (e.g. see Villandsen et al., 2023; J Neurochem. **165**:76-94. doi: 10.1111/jnc.15754.) but this is outside the scope of the present manuscript and will be the subject of further investigations.

The analysis is very vague, please include more detail: Line 507 – what is this semi-automated bioinformatics workflow, can you provide brief description?

Author’s answer: As suggested by the reviewer, we have modified line 558-565: “The proteins identified in FI, FIII, FIV1-4 were characterized by an integrated databases analysis divided into four main steps: a) the selection of identified proteins form the comparison with protein databases (The Human Protein Atlas, proteinatlas.org; and HUPO, hupo.org; Plasma Protein database, plasmaproteomedatabase.org) to select plasma proteins, b) the association of proteins with relevant human diseases, c) the existence of relevant animal models and other relevant tools (antibodies, assays, etc.) and d) the evaluation of technical feasibility for the development of the identified targets into therapeutic approach.” The whole process is described in detail in lines from 558 to 565 in the Materials and Methods section. For the benefit of the reviewer, we briefly summarize here the process.

Once proteins of each fraction were identified by using Proteome Discover 2.5 (ThermoFisher Scientific) they underwent to a bioinformatics pipeline to achieve these four main steps:

1. Selection of proteins for a comparison with the major plasma proteomic databases: Plasma Protein database (plasmaproteomedatabase.org), the Human Protein Atlas (proteinatlas.org) and HUPO (hupo.org). For completeness, plasma protein concentrations reported in Human protein Atlas (Immunoassay and/or Mass Spectrometry) were also included.
2. Disease-association cross-analysis by assessing pathology and human genetics databases, and specifically: DisGeNet (disgenet.org), MalaCards (malacards.org), GeneCards (genecards.org), HumanMine (humanmine.org), eDGAR (edgar.biocomp.unibo.it), DAVID (david.ncifcrf.gov). Additional data on the incidence of relevant diseases, their inheritance and age of onset was gathered from OMIN (omim.org) and Orphanet (www.orphanet.net).
3. For each protein, the identification of relevant animal models was performed by interrogating the International Mouse Phenotyping Consortium (mousephenotype.org) and Mouse Genome (informatics.jax.org) databases.
4. The resulting subset of proteins was then compared with literature and patent databases (Pubmed and Espacenet) to assess technical feasibility for development of the identified targets through the various therapeutic modalities.

Further, proteomics sample prep is not clear – the samples were run through filtration unit prior or after reduction and alkylation? Please make it clear, also what speed in the centrifuge was the filtration unit spun for? Why were filtration units used, to remove detergents, if that is the case, then why was the washes with UREA introduced? How many fractions were generated from the solid-phase extraction cartridges, what was the condition?

Author's answer: The method used for proteomic sample preparation is the Filter aided sample preparation (FASP) developed by Matthias Mann, in common use as reported in a broad range of proteomics papers (e.g. see Wiśniewski JR, et al. 2009, Nat Methods **6**:359-362. doi: 10.1038/nmeth.1322). Filtration was performed prior to reduction and alkylation. This FASP method was selected in order to remove the SDS detergent from the samples. In particular, at the beginning of the preparation the samples were loaded on 30-kDa cutoff molecular weight filters and then reduction and alkylation were performed. The filters were centrifuge at 13,800 g. The UREA was used in order to clean-up the sample as suggested from the FASP method (Zhang Z, et al. 2020, Anal Chem. **92**:5554-5560. doi: 10.1021/acs.analchem.0c00470). The tryptic peptides obtained from digestion of the plasma fractions were fractionated using solid phase extraction cartridges HRP (Reversed Phase Polymeric) C-18 SOLATM (by Thermo Fisher Scientific) in 8 different fractions using High-pH reversed-phase chromatography eluting peptides using buffers with 0.1% triethylamine and an increasing concentration of organic solvent (acetonitrile). We have modified the Material and Methods section in order to clarify that the process was performed prior to reduction and alkylation (Lines 531-548).

What is the intensity of these proteins identified from your proteomics data – how many peptides etc, please include this in your SM1.

Author's answer: The coverage, the number of peptides, the number of unique peptides and the average of the abundance based on peak intensity have now been included in SM1 for each protein of every fraction, as suggested by the reviewer.

5. What are these known proteases that cleave CP protein resulting in different band sizes on the western blot? Can you include this in your discussion? Further, why are those bands also present in the CP alone sample, it should not have any other contaminant proteins, please discuss.

Author's answer: We thank the reviewer for point this out, which allowed us to extend our considerations on the partial proteolysis associated with kCP, the starting material from which kCP was purified (FrIV₁₋₄) and commercially available CP. CP is a substrate of different proteases, including Trypsin-like proteases like Thrombin to cite the most commonly reported. Thrombin is reported as the protease that causes the limited proteolysis of CP observed both *in vitro*, during purification and *in vivo* (Vasyliov 2019- Biometals **32**:195-210. doi: 10.1007/s10534-019-00189-1). Commercially available CP (used in this study and in Zanardi et al., 2018) presents evidence of proteolysis relative to plasma (Fig. SM4.2), indicative of partial proteolysis resulting from plasma fractionation and/or subsequent CP purification. As for Kedrion's samples, we can also detect increased proteolysis in the starting material (unused industrial plasma fractionation intermediates) relative to whole plasma (Fig. SM4.2A, Fig. 3A1). This partial proteolysis does not increase with subsequent purification of kCP (Fig. SM4.2A and B), indicative that proteolysis is likely the result of events occurring during the plasma fractionation process (e.g. the result of physical processing or the activity of proteases which co-fractionate with CP). Our data with kCP as well as the previous study published using the commercially available CP demonstrate that this partial proteolysis does not influence efficacy in the aceruloplasminemia mouse model. We therefore believe that the observed partial degradation pattern is likely due to events occurring mostly during plasma fractionation, rather than kCP purification, though some evidence of partial proteolysis is observed also in whole human plasma (and this likely due to the activity of endogenous proteases). The bands corresponding to both full length and (to a lesser extent) fragmented CP are detected by the anti-CP antibody in a Western blot in all samples (the starting material as well as the purified CP, e.g. "CP alone" sample) as CP fragments co-purify with full length CP during the purification process. As for other contaminants, plasma-derived CP is by its nature (as are all other licensed and commercialized plasma-derived products) a protein concentrate, consisting of a majority of CP and a smaller proportion of other proteins which co-purify with CP (which we have characterized and reported in the manuscript in Fig. 3). We have included a sentence in the Results (rather than in the Discussion) section to explain this (lines 144-151).

6. For your Figures, some of them had detailed stats, while others were missing stats, see below for a good example from one your figures, please apply this to all. "Data are presented as mean \pm SEM; each dot corresponds to one animal (N= 20 and N= 40 for cpKO at 6 months). Statistical P values were evaluated by one-way ANOVA (An) followed by Tukey's post-test analysis (panels B,C,G,H) or Kruskal-Wallis (K-W) test followed by either Dunn's (panel A) or uncorrected Dunn's (panels D,E) post-test analysis."

Author's answer: The statistical details have been implemented in the legend of figures 3 and 4, as suggested.

7. Why weren't 33 parameters used to make Fig 8A, why were those parameters excluded, it is not clear, please discuss.

Author's answer: Figure 8A refers to mice at 6 months of age and all parameters measured were accordingly included in the analysis (see last paragraph of the Results section). The 33 parameters used to generate panel 8B included all the phenotype evaluations performed on organs collected at the end of the treatment, thus not measurable at 6 months in same group of animals (see same section as above).

Minor comments to be addressed

1. Fig1b – you can see a dark faint line around the figure, please fix

Author's answer: We thank the referee for this comment, but we don't see this in our original version. This artefact is likely automatically generated by the CommBiol website during the conversion to pdf file, nevertheless this has now been fixed (see Fig1b in the revised manuscript). In any case, a high-resolution image file can be sent if the manuscript will be accepted.

2. Explain what GFAP and IBA1 is for non-expert reader?

Author's answer: the extended definition for GFAP and IBA1 molecules and their role as marker for astrocytes and microglial cells, respectively, have been introduced in the results section where abbreviations appear for the first time (lines 254-255).

3. The stats for ACP are in the discussion, I would prefer it in the introduction, can you can please move this?

Author's answer: the sentence referring to ACP prevalence has been moved in the introduction (lines 71-72)

4. Why does lipid and iron accumulation correlate in the liver? Please discuss.

Author's answer: A dysregulated interaction of iron and lipid metabolisms has been reported in several metabolic diseases and is characterized by altered levels of key players in the regulation of iron and lipid homeostasis (e.g. hepcidin in liver and adipokines in adipose tissue)(Barisani D, et al. Hepcidin and iron-related gene expression in subjects with Dysmetabolic Hepatic Iron Overload. J Hepatol 2008;49:123–33. doi.org/10.1016/j.jhep.2008.03.011.; Gabrielsen JS, et al. Adipocyte iron regulates adiponectin and insulin sensitivity. J Clin Invest 2012;122:3529–40. doi.org/10.1172/JCI44421; Gao Y, et al. Adipocyte iron regulates leptin and food intake. J Clin Invest 2015;125:3681–91. doi.org/10.1172/JCI8186; Yamamoto K, et al. Interplay of adipocyte and hepatocyte: Leptin upregulates hepcidin. Biochem Biophys Res Commun 2018;495:1548–54. doi.org/10.1016/j.bbrc.2017.11.103.). Of interest are the observations that reduced levels of CP have been reported in some metabolic diseases characterized by liver fat accumulation and/or iron overload like non-alcoholic fatty liver disease and non-alcoholic steatohepatitis patients (Nobili V, et al. Levels of serum ceruloplasmin associate with pediatric nonalcoholic fatty liver disease. J Pediatr Gastroenterol Nutr 2013;56:370–5.

doi.org/10.1097/MPG.0b013e31827aced4; El-Rayah E-GA, et al. Both α -1-antitrypsin Z phenotypes and low caeruloplasmin levels are over-represented in alcohol and nonalcoholic fatty liver disease cirrhotic patients undergoing liver transplant in Ireland. *Eur J Gastroenterol Hepatol* 2018;30:364–7. doi.org/10.1097/MEG.0000000000001056; Aigner E, et al. Copper availability contributes to iron perturbations in human nonalcoholic fatty liver disease. *Gastroenterology* 2008;135:680–8. doi.org/10.1053/j.gastro.2008.04.007; Wang Q, et al. A Novel Non-Invasive Approach Based on Serum Ceruloplasmin for Identifying Non-Alcoholic Steatohepatitis Patients in the Non-Diabetic Population. *Front Med* 2022;9:900794. doi.org/10.3389/fmed.2022.900794; France M, et al. Liver Fat Measured by MR Spectroscopy: Estimate of Imprecision and Relationship with Serum Glycerol, Caeruloplasmin and Non-Esterified Fatty Acids. *Int J Mol Sci* 2016;17:E1089. doi.org/10.3390/ijms17071089). At the cellular/biochemical level, there is a convergence between the iron and lipid pathways within the mitochondria. This occurs because changes in iron balance and in the pool of labile iron can influence the activity of aconitase, which in turn affects the Krebs cycle and the levels of acetyl-CoA. These levels play a crucial role in regulating the metabolic pathways involved in FA, TG, and cholesterol synthesis and degradation (Rockfield S, et al. *Links Between Iron and Lipids: Implications in Some Major Human Diseases. Pharmaceuticals* 2018;11:E113. doi.org/10.3390/ph11040113.). Of note, mitochondrial dysfunction has been reported in ACP patients, at least in the brain (Miyajima H, et al. *Increased lipid peroxidation and mitochondrial dysfunction in aceruloplasminemia brains. Blood Cells Mol Dis* 2002;29:433–8. doi.org/10.1006/bcmd.2002.0561), which in turn suggests an opportunity to further investigate this topic in the CP-deficient mouse model of ACP. However, since we were mainly interested in the efficacy of the CP PRT, at both CNS and systemic level, in our manuscript we are just focussing on mitigation of the key lipid dysmetabolism phenotypes reported in ACP patients and in the CP-deficient mouse model. We avoided to address this in the discussion as we believed to be out of scope for the present manuscript and the manuscript already covers several key therapeutic benefits of CP PRT in ACP, though we agree it is an interesting aspect worthy of further investigation.

5. Line 427 – missing a word

Author's answer: fixed (now line 473)

6. Line 461 should be are

Author's answer: done (now line 507)

7. How were samples prepared for ICP-MS, a brief description is required?

Author's answer: the description of samples preparation has been introduced in the Materials and Methods section. (lines 727-728).

8. Why does it jump from SM3 to SM6, why isn't it just SM4?

Author's answer: We have difficulties in understanding this point. The Supplementary Materials is just a unique pdf file in which the information go through SM1 to SM6 and where notes indicate that SM1, SM2, SM3 and SM6 are provided as separated excel files. Within the main text the different supplementary materials are progressively quoted from SM1 to SM6 including SM4 and SM5 (i.e. SM1, SM2 and SM3 page 5 line 90, 100 and 106; SM4 page 6 line 137; SM5 page 7 line 164; SM6 page 12 line 327; of the new version).

9. Line 90 needs to be re-written, what do you mean “associated with a known Homo sapiens gene”?

Author’s answer: As suggested we have modified this sentence (line 91): “Within these, 308 human plasma proteins were found, of which...”

10. In Fig6, what does AN= MEANS?

Author’s answer: As indicated in the figure 6 legend, “An” means ANOVA and indicates the overall ANOVA p value calculated for the 3 groups. In addition in the graphs, the p values of post hoc test are also reported for direct comparison of group pairs.

11. Fig5, H and I are switched around. Fig 3, A2 and 3 are switched around. Please ensure the correct figure corresponds to the correct legend and also within text.

Author’s answer: Fixed by switching around in the figure legends.

12. Fig SM4.2 – why can’t you detect the proteolytic versions of CP in the plasma of mouse?

Author’s answer: Commercially available CP (used in this study and in Zanardi et al., 2018) presents evidence of proteolysis relative to plasma (Fig. SM4.2), indicative of partial proteolysis resulting from plasma fractionation and/or subsequent CP purification. As for Kedrion’s samples, we can also detect increased proteolysis in the starting material (unused industrial plasma fractionation intermediates) relative to whole plasma (Fig. SM4.2A, Fig. 3A1). This partial proteolysis does not increase with subsequent purification of kCP (Fig. SM4.2A and B), indicative that proteolysis is likely the result of the plasma fractionation process (e.g. the result of physical processing or the activity of proteases which co-fractionate with CP). The difference between plasma and industrial plasma fractions explains why the partial proteolysis is not observed in freshly collected mouse plasma as this has not been subjected to any processing. Our data with kCP as well as the previous study published using the commercially available CP demonstrate that this partial proteolysis does not influence efficacy in the aceruloplasminemia mouse model.

13. FigSM5.1, there seems to be a slight decrease in body weight at 5ug/g, can you explain this

Author’s answer: We believe that the slight decrease observed is within physiological variance and that, due the limited number of animals used in the experiment n= 3, resulted in an apparent decrease. Indeed, was not correlated with the dosage of CP administered. Moreover, the occasionality/irrelevance of this body weight decrease is also supported by the new data, introduced in the manuscript to answer to this Referee’s point 3, on toxicity analysis performed on WT mice treated with 5ug/g CP for 4 months. In this larger group of animals (n= 10) compared with the experimental group of WT mice treated with saline in the study (n= 20) no differences in body weight can be observed along the treatment (see SM5.2).

14. Where is the raw proteomics data deposited?

Author’s answer: The mass spectrometry proteomics data have been deposited to the ProteomeXchange Consortium via the PRIDE partner repository with the dataset identifier PXD046234. Data are available via ProteomeXchange with identifier PXD046234. This information is now indicated in the notes “Data and materials availability”.

15. It's not clear how long mice were treated with kCP, can you make it clear prior to every results section.

Author's answer: The requested information has been added in all relevant paragraphs of the Results section (e.g. line 222; line 298; line 324-325).

16. Line 196 – can you state this significant difference in number, this applies to most situation in the results section, where you have mentioned that there is a significant difference, it would be good to see what that number is and the associated p-value.

Author's answer: In order not to weigh down the main manuscript text, which is dense with information, individual data and statistical significance values are present directly in the Figures. We believe this provides the necessary information while maintaining the readability of the main text, but if this Reviewer believes that it is essential we can introduce numbers throughout the text.

Reviewer #2 (Remarks to the Author):

Zanardi et al. describe the proteomic characterisation of unused plasma fractions, and the isolation and utilization of ceruloplasmin from these fractions. The proteomic analysis undertaken appears to be of high quality, however further description of the quantitative analyses carried out is needed.

Author's answer: We thank this Reviewer for considering our proteomics analysis of high quality. A more detailed description of the proteomics characterization was correctly requested by more than one reviewer and the SM1 section was integrated accordingly, please see the Materials and Methods section (lines 531-548). Additionally, to provide more details (including intensity evaluation) on our results, we have added information on the coverage, the number of peptides, the number of unique peptides for each protein of every fraction in SM1 file. We believe we addressed this Referees' points through an in depth revision of the manuscript (modified/new text highlighted in yellow, with reference to relevant lines in the text).

Minor comments:

1. The authors describe the relative quantification of CP in the fraction FIV1-4 using mass spectrometry, however the relevant methods (pg. 19) do not describe how the quantitation was done. Table SM3 does not provide more context in this regard. Was the quantitation done through ProteomeDiscoverer using a label-free method? A more detailed description is needed in the methods to describe how the quantitation, and not just identification, of proteins in fraction FIV1-4 was achieved.

Author's answer: We thank the Reviewer for the suggestion. In the first sheet of SM3, we have added the following sentence describing the quantification method (highlighted in yellow): "Protein identification and quantification were performed on the whole data set using t Proteome Discoverer 2.5. A given protein was considered as "identified" when a valid

MS2 spectrum was available for at least one of the peptides belonging to that protein. PD provided for each quantified protein the height of the most abundant peak at the apex of the chromatographic profile (“intensity”). Quantitative values were subjected to a normalization step, based on the total peptide intensity of the samples (Palomba, A., et al., Comparative evaluation of MaxQuant and proteome discoverer MS1-based protein quantification tools. Journal of proteome research, 20(7), 3497-3507. 2021) from which the % of each abundance protein is obtained”.

2. Relatedly, the authors note that in their proteomic analyses they estimate 98% purity of CP, as opposed to 80% purity by ELISA (pg. 6, line 138). I note that the authors use 30-kDa cutoff molecular weight filters in their sample preparation for mass spectrometry. This raises the question as to whether proteins smaller than 30 kDa are being lost in the sample processing, and whether this is skewing the relative abundance of CP in the sample? The authors should consider this in the interpretation of their results.

Author’s answer: The method used for proteomic sample preparation is the Filter aided sample preparation (FASP) developed by Matthias Mann, as reported in a broad range of proteomics papers (Wiśniewski JR, et al., Universal sample preparation method for proteome analysis. Nat Methods. 2009 May;6(5):359-62. doi: 10.1038/nmeth.1322. Epub 2009 Apr 19. PMID: 19377485.). In particular, based on several papers (e.g. see Zhang Z, et al., Miniaturized Filter-Aided Sample Preparation (MICRO-FASP) Method for High Throughput, Ultrasensitive Proteomics Sample Preparation Reveals Proteome Asymmetry in Xenopus laevis Embryos. Anal Chem. 2020 Apr 7;92(7):5554-5560. doi: 10.1021/acs.analchem.0c00470. Epub 2020 Mar 12. PMID: 32125139; PMCID: PMC7931810) we have optimized the initial protocol by Mann’s group by replacing the 10kDa cutoff molecular weight filters with the 30-kDa cutoff molecular weight filters thus improving the number of identified proteins. *Therefore, as reported in the literature the use of 30-kDa cutoff molecular weight filters didn’t lead to the loss of proteins MW<30kDa since the proteins bind to SDS forming micelles (MW>30kDa).* We have modified the Material and Methods section in order to clarify the process, please see lines 531-548.

3. The authors mention that peptides were fractionated offline with SPE (pg. 18, line 492). More details are needed here. What specific cartridges were used (what chemistry), how was the fractionation done, and how many fractions were collected/analyzed?

Author’s answer: The tryptic peptides obtained from digestion of the plasma fractions were fractionated using solid phase extraction cartridges HRP (Reversed Phase Polymeric) C-18 SOLA TM (by Thermo Fisher Scientific) in 8 different fractions using High-pH reversed-phase chromatography eluting peptides using buffers with 0.1% triethylamine and an increasing concentration of organic solvent (acetonitrile). We have added a sentence in the Materials and Methods section to clarify this as requested by the reviewer see lines 531-548.

Reviewer #3 (Remarks to the Author):

This manuscript (COMMSBIO-23-3004-T) is the second proof-of-concept study from Dr. Alessio’s group testing whether ceruloplasmin (CP) isolated from human plasma can be used as a therapeutic agent for patients with Aceruloplasminemia (ACP). ACP is a great candidate

for PRT approach because it is a monogenic disease, and CP works extracellularly. In summary, these two studies are rather similar besides the source of CP with potentially a new extraction method and duration of treatment. However, this study can be a valuable addition to the field of rare diseases because they also found other diseases (interestingly all iron metabolism-related) that might be candidates for the unused plasma-driven therapeutics approach. I recommend this manuscript to be accepted for publication with some revision. Detailed critique below.

Author comments. We thank the Reviewer for her/his positive evaluation of our work and for the precious suggestions how to improve the quality of the manuscript. We believe we addressed this Referees' points through an in depth revision of the manuscript (modified/new text **highlighted in yellow**, with reference to relevant lines in the text).

In particular, the first point raised by this Referee gives us the opportunity to address one of the key aspects raised by the Editor. We agree that this is the second proof-of-concept of the feasibility of PRT using CP purified from plasma, however we believe that our manuscript presents several key differentiators and extensions from the previous study. Aside from confirmatory evidence (nevertheless important in its own right), the present manuscript extends previous data on the therapeutic efficacy of CP PRT with these novel findings:

- A) The demonstration through radiolabelling experiments that CP reaches the brain, lines 211-216
- B) The recovery of neurodegeneration in both cerebellum and hippocampus (the latter, never considered before in ACP mice, is an area of key importance to cognitive and behavioural deficits in aceruloplasminemia patients), included in lines 226-291 (the beam walking test analysis has also not been previously reported)
- C) The mitigation of neuroinflammation in both cerebellum and hippocampus (the observation of neuroinflammation in ACP mice and its mitigation in both brain areas is also entirely novel), included in lines 252-291
- D) The recovery of defective erythropoiesis, lines 315-321

In the Discussion section we have highlighted most of these novel findings in lines 433-435. More generally, the demonstration that CP replacement therapy is possible using a CP protein purified from an industrial plasma fractionation waste intermediate represents a significant improvement from several points of view. First, in terms of optimizing plasma use (a precious resource resulting from the generosity of donors), has a high ethical value which the entire plasma Industry is striving to achieve. Second, this ethical value is further strengthened by the possibility of recycling an industrial waste product, currently disposed of as special waste, into the manufacturing process, which represents a significant improvement in the context of our society's current effort to be "green" and reduce/recycle industrial waste. Third, the use of an industrial waste as starting material for the manufacturing of a therapy for an ultra-rare disease significantly improves the financials in terms of manufacturing costs and therefore of therapy cost for patients and Health Systems. Last, aside from CP there are many other proteins present in unused intermediates which represent therapeutically useful candidates or even therapies currently on the market, and their manufacturing from waste fractions could reduce the dependence of some geographies (e.g. >35% of Europe's plasma supply is of US origin) from the necessity to import plasma. This has also been discussed in the Discussion section, e.g. see lines 344-362.

Reviewer:

1) Please strengthen the immune response data. Many PRT or ERT (enzyme replacement therapy) induce immune response to foreign protein, therefore, immune response to kCP is one of the key pieces of data if this type of therapeutic approach goes to clinic. In this manuscript, the data is only available in the supplemental data section (SM 5.4) and there is barely any discussion in the manuscript.

Author's answer: Based on this Reviewer's comment, we have now moved data related to the induction of an anti-CP immune response in our PRT from Supplemental Materials to the Results section of the main text (lines 197-208) and as part of Figure 4 (new panels C,D and E). Moreover, a comment on the general issue of immune response induction in ERT has been introduced in the Discussion section (lines 410-419) (see also answer to Reviewer #1 point 1).

2) What if the anti-CP antibody interferes with CP/FPN interaction? CP is a ferroxidase, therefore testing the ferroxidase activity in the presence of anti-CP antibody is quite logical in Fig SM 5.4-C. However, CP as a ferroxidase is involved in iron efflux partnering with the sole iron efflux protein FPN. If this anti-CP antibody interferes with CP/FPN interaction, it will diminish cell's iron efflux capacity and cause ACP-like situation. This possibility needs to be addressed before we can deem this anti-CP antibody non-problematic.

Author's answer: The ferroxidase activity of CP/GPI-CP has been reported to be necessary for the stabilization of FPN on the cell membrane e for the maintenance of its iron-exporter activity (Jeong et al., 2003, J Biol Chem; De Domenico et al., 2007, EMBO J; Kono et al. 2010, Biochem Biophys Acta). Given that the interaction between CP and FPN has been demonstrated to be a functional interaction and not a structural/physical interaction (Jeong et al., 2003, J Biol Chem; De Domenico et al., 2007, EMBO J; di Patti et al., 2009, J Biol Chem), and that no evidence has so far been obtained for a direct physical interaction between CP and FPN (e.g. Musci et al., 2014, World J Biol Chem. 2014 May 26; 5(2): 204–215), it is highly unlikely that the **non-neutralizing** anti-CP antibodies raised during CP replacement therapy in our study might result in impairment of FPN function. This conclusion is strongly supported by the mitigation of the many phenotypes resulting from CP replacement therapy in ACP mice. These include reduction of iron accumulation in brain and in particular in the choroid plexus epithelial cells; reduction of iron accumulation in liver; more indirectly, amelioration of neurological, neurodegenerative and neuroinflammatory phenotypes which in this genetic model of ACP are directly linked to iron metabolism dysfunction as well as the amelioration of erythropoietic phenotype. These effects of CP replacement therapy are directly linked with amelioration of iron metabolism, and would have not been possible if the anti-CP antibodies had significantly impacted FPN function. On the contrary, these effects are consistent with the non-neutralizing nature of the anti-CP antibodies, which leaves CP ferroxidase activity intact and therefore able to interact **functionally** with FPN. It may however explain some of the variability between mice in the treatment groups (e.g. see Fig. 5G), as the immune response to CP replacement therapy in individual mice might be somewhat variable. However, we could not investigate anti-CP antibodies from individual CP PRT treated mice (and instead analyzed pooled samples from each treatment group) for practical reasons (amount of IgG which can be purified from the small volume of serum from an individual mouse).

3) ⁶⁴Cu-CP distribution experiment is quite important. As mentioned above, ACP is probably one of the most logical targets for PRT because the secreted form of CP is abundant in plasma and produced by liver. However, CNS expresses GPI-anchored form of CP and isolated behind BBB. Although expected from the improvement of CNS phenotypes in their 2018 paper, it is quite cool to see that the radio-labeled CP was detected directly inside brain. In my opinion, this is a piece of very important data and yet hidden away in SM 5.3.

Author's answer: As suggested the result of the ⁶⁴Cu-CP distribution analysis in the brain has been moved to the Results section (lines 211-216; Fig. 5A) of the main text of the manuscript and quoted in the Discussion section (line 402; lines 427-430). Whereas the results of the ⁶⁴Cu-CP accumulation in blood and plasma have been maintained as supplementary materials, since they are only supportive of the results shown in figure 3A. The graph related to ⁶⁴Cu-CP distribution in the brain has been included in Fig. 5 (as new panel A) and methods for CP labelling and administration have been accordingly included in the Materials and Methods section of the revised version of the manuscript (lines 666-697).

4) Discussion on possibility of secreted, BBB-crossed kCP to fill in the role of GPI-CP. As mentioned above, CNS expresses alternatively spliced form of CP, and it seems not clear why brain prefers anchored protein unlike plasma. It is quite interesting that human plasma-isolated secreted CP can alleviate neurological phenotypes in Cp null mice. A section in Discussion regarding this point would be beneficial to the field.

Author's answer: The function of GPI-anchored is still poorly understood (for instance there is no specific mouse KO model where secreted CP is maintained while abolishing expression of the secreted form). The evidence from this manuscript and other published work (Zanardi et al., 2018) clearly indicates that, in the absence of any CP (secreted or GPI-anchored) systemically administered CP enters the brain and mitigates iron-related neurological, neurodegenerative and neuroinflammatory phenotypes. In neuronal cell models, the administration of exogenous CP is able to maintain FPN functionality for iron export (Olivieri et al., 2011; J Neurosci. 2011 Dec 14;31(50):18568-77. doi: 10.1523/JNEUROSCI.3768-11.2011). Overall, therefore, while expression of a GPI-anchored form of CP is reported, its role in the brain (and, of relevance to the present manuscript, in ACP phenotypes) is unclear. CP is present in CSF, and is therefore available to the CNS, from release of GPI-anchored CP from astrocytes, from secretion of CP from choroid plexus epithelial cells or from transport from plasma (this latter particularly in neurodegenerative contexts). We have added a sentence in the discussion to mention this aspect (line 440-444) ("However, the physiological role for this isoform in vivo remains unclear, and the ability of CP PRT to mitigate phenotypes in CP knock-out mice (Zanardi et al., 2018; this manuscript) indicates that its lack in the CNS can be significantly complemented by systemically delivered (or secreted) CP.").

REVIEWERS' COMMENTS:

Reviewer #1 (Remarks to the Author):

The authors have addressed the concerns of the prior review, no further concerns have been raised.

Reviewer #2 (Remarks to the Author):

The authors have addressed all my concerns.

Reviewer #3 (Remarks to the Author):

In this manuscript (COMMSBIO-23-3004-T), Dr. Alessio's group showed

- 1) They can purify functional proteins from human plasma otherwise will be discarded. These proteins can be a potentially therapeutic agent for a number of monogenic diseases and here they tested the concept on a mouse model for aceruloplasminemia (aCP).
- 2) Purified human plasma ceruloplasmin (kCP) retained enzymatic activity, alleviated phenotypes in liver and brain (two main symptomatic areas in this mouse) and was able to cross blood-brain-barrier.
- 3) kCP also showed neuroprotective effect and prevented glial activation in this mouse model.
- 4) These antibodies generated against kCP are not interfering with ceruloplasmin enzyme activity or causing protein degradation, which are two common concerns of ERT (enzyme replacement therapy).

As authors explained, only a certain number of human plasma proteins are purified and utilized through pharmaceutical manufacturing and the rest is discarded. This manuscript suggested a novel concept of purifying ERT candidate proteins from this discarded fraction of human plasma ("upcycling"). Besides aCP, they also identified three other rare diseases that can potentially benefit from this approach.

In their revised version of the manuscript, they addressed each reviewer's critique point-by-point and updated their manuscript. Therefore, in this reviewer's opinion, this manuscript is a cool pre-clinical, proof-of-concept paper to explore a new protein source for human ERT trials.